# An allosteric Sec61 inhibitor traps nascent transmembrane helices at the lateral gate

Andrew L MacKinnon[1†], Ville O Paavilainen[1†], Ajay Sharma[2], Ramanujan S Hegde[2‡], Jack Taunton[1,3]*

[1]Department of Cellular and Molecular Pharmacology, University of California, San Francisco, San Francisco, United States; [2]Cell Biology and Metabolism Program, National Institutes of Child Health and Human Development, National Institutes of Health, Bethesda, United States; [3]Howard Hughes Medical Institute, University of California, San Francisco, San Francisco, United States

**Abstract** Membrane protein biogenesis requires the coordinated movement of hydrophobic transmembrane domains (TMD) from the cytosolic vestibule of the Sec61 channel into the lipid bilayer. Molecular insight into TMD integration has been hampered by the difficulty of characterizing intermediates during this intrinsically dynamic process. In this study, we show that cotransin, a substrate-selective Sec61 inhibitor, traps nascent TMDs in the cytosolic vestibule, permitting detailed interrogation of an early pre-integration intermediate. Site-specific crosslinking revealed the pre-integrated TMD docked to Sec61 near the cytosolic tip of the lateral gate. Escape from cotransin-arrest depends not only on cotransin concentration, but also on the biophysical properties of the TMD. Genetic selection of cotransin-resistant cancer cells uncovered multiple mutations clustered near the lumenal plug of Sec61α, thus revealing cotransin's likely site of action. Our results suggest that TMD/lateral gate interactions facilitate TMD transfer into the membrane, a process that is allosterically modulated by cotransin binding to the plug.

*For correspondence: jack.taunton@ucsf.edu

†These authors contributed equally to this work

‡Present address: MRC Laboratory of Molecular Biology, Cambridge, United Kingdom

Competing interests: The authors declare that no competing interests exist.

## Introduction

Most eukaryotic membrane proteins are cotranslationally integrated into the endoplasmic reticulum (ER) membrane (*Shao and Hegde, 2011*). This process begins when the first hydrophobic segment of a nascent membrane protein, often a transmembrane domain (TMD), emerges from a translating ribosome and is recognized by the signal recognition particle (SRP). The SRP system (*Akopian et al., 2013*) targets the ribosome-nascent polypeptide complex (RNC) to the ER membrane and transfers it to a translocation channel, or translocon, the central component of which is the Sec61 complex (*Osborne et al., 2005*). Sec61 binds the ribosome near the polypeptide exit tunnel and mediates both cotranslational translocation and TMD integration. This means that the Sec61 channel opens not only toward the ER lumen, but also laterally toward the lipid bilayer. While lateral TMD release through the Sec61 channel was appreciated long ago (*Martoglio et al., 1995*; *Do et al., 1996*; *Mothes et al., 1997*), the precise mechanism of this crucial step in membrane protein biogenesis remains unclear.

During integration into the membrane, TMDs must move from the aqueous pore of the Sec61 channel into the surrounding lipid bilayer. Several models have been proposed to explain how this occurs. In one model, the TMD is thought to equilibrate between the aqueous Sec61 channel and the lipid bilayer (*Heinrich et al., 2000*; *Hessa et al., 2007*; *Ojemalm et al., 2011*). This process would be facilitated by an intrinsically dynamic lateral gate in Sec61 that allows the polypeptide segment inside the channel to reversibly sample the membrane. A second model proposes that TMD integration is facilitated by direct interactions with Sec61. This idea is supported by the analysis of stalled RNCs of membrane proteins at different lengths. Not only does the TMD of such stalled RNCs crosslink to Sec61,

**eLife digest** Cells are surrounded by a plasma membrane that acts like a barrier around the cell—keeping the cell's boundaries distinct from surrounding cells and helping to regulate the contents of the cell. This plasma membrane is made up mostly of two layers of fatty molecules, and is also studded with proteins. Some of these membrane proteins act as channels that allow nutrients and other chemicals to enter and leave the cell, while others allow the cell to communicate with other cells and the outside environment.

Like all proteins, membrane proteins are chains of amino acids that are linked together by a molecular machine called a ribosome. The ribosomes that make membrane proteins are located on the outside of a membrane-enclosed compartment within the cell called the endoplasmic reticulum. To eventually become embedded within a membrane, a new protein must—at the same time as it is being built—enter a channel within the membrane of the endoplasmic reticulum. The newly synthesized protein chain enters this channel, called Sec61, via an entrance near the ribosome and then threads its way toward the inside of the endoplasmic reticulum. However, there is also a 'side-gate' in Sec61 that allows specific segments the new protein to escape the channel and become embedded within the membrane. From here, the membrane protein can be trafficked to other destinations within the cell, including the plasma membrane. However, how the newly forming protein chain passes through the side-gate of Sec61 is not well understood.

Now MacKinnon, Paavilainen et al. have used a small molecule called cotransin—which is known to interfere with the passage of proteins through Sec61—to observe the interactions between the Sec61 channel and the new protein. Cotransin appears to trap the new protein chain within the Sec61 channel by essentially 'locking' the side-gate.

MacKinnon, Paavilainen et al. observed that the trapped protein interacts with the inside of the channel at the end closest to the ribosome—which is the likely location of the side-gate. In contrast, cotransin likely binds at the other end of the channel, to a piece of Sec61 that serves to plug the exit into the endoplasmic reticulum; and this plug is directly connected to the side-gate. By preventing the plug from moving out of the way, cotransin can somehow stop the new protein from passing through the side-gate. However, MacKinnon, Paavilainen et al. did find that some membrane proteins with certain physical and chemical properties could get through the gate, despite the presence of cotransin. The next challenge is to resolve exactly how interactions between cotransin and the Sec61 plug can block the escape of new proteins into the membrane.

but it also appears to be oriented in a specific manner, as judged by asymmetric crosslinks to adjacent TMD residues (*McCormick et al., 2003*; *Sadlish et al., 2005*). Molecular dynamics simulations further suggest that lateral gating is facilitated by the TMD and that TMD movement into the lipid bilayer is a kinetically irreversible event (*Zhang and Miller, 2010*, *2012*). Analysis of cotranslational integration in yeast has also revealed a kinetic barrier to TMD insertion (*Cheng and Gilmore, 2006*; *Trueman et al., 2011*).

Structural studies of the Sec61 complex have provided important clues to the mechanism of TMD integration. The crystal structure of the archaeal complex revealed that SecY (homologous to eukaryotic Sec61α) comprises a compact bundle of 10 transmembrane helices (TM1–10) arranged in a pseudo-twofold symmetric structure, with TM1–5 and 6–10 forming two halves of a 'clamshell' (*Van den Berg et al., 2004*). The interior of the clamshell forms an hourglass-shaped pore in the membrane, the center of which is occluded by a short α-helix termed the plug. The front of the clamshell, at the interface between TM2/3 and TM7/8, forms a lateral gate that, when opened, could provide direct access to the lipid bilayer from the central pore.

Translocation across the membrane requires movement of the plug to open the channel toward the ER lumen. This is thought to occur when the first TMD (or signal sequence) of an RNC intercalates between the lateral gate helices of Sec61α. When coupled with ongoing translation, this event would position the nascent polypeptide within the channel and provide access of the hydrophobic TMD to the lipid bilayer. This idea is supported by the observation that signal peptides can be crosslinked to Sec61α TM2 and TM7 (*Plath et al., 1998*; *Wang et al., 2004*) and that mutations in the lateral gate can influence TMD integration (*Junne et al., 2010*; *Trueman et al., 2012*). While crystal structures of prokaryotic Sec61 have revealed the lateral gate in a continuum of partially open states (*Tsukazaki*

*et al., 2008*; *Zimmer et al., 2008*; *Egea and Stroud, 2010*), how the gate transitions between open and closed conformations, and especially the role of the nascent TMD in this process, remains unclear. A major obstacle to understanding cotranslational TMD integration has been the inability to stabilize and interrogate discrete pre-integrated intermediates during this highly dynamic process.

We and others previously described cotransins, a class of cyclic peptides that bind Sec61α and potently inhibit cotranslational translocation of a subset of secretory and membrane proteins (*Besemer et al., 2005*; *Garrison et al., 2005*; *MacKinnon et al., 2007*). We reasoned that because Sec61 represents the final destination of a nascent TMD prior to its integration into the membrane, small-molecule modulators like cotransin could shed light on this poorly understood process. In this study, we have investigated the mechanism by which cotransin inhibits membrane protein insertion. We demonstrate that cotransin stabilizes an ephemeral pre-integrated intermediate in which the TMD docks against the cytosolic tip of the Sec61 lateral gate. Progression through this cotransin-arrested stage was found to depend on TMD hydrophobicity, helical propensity, and charge distribution, features that contribute to cotransin's ability to discriminate among different substrates. Using genetic selection in human cancer cells, we identified multiple point mutations in the lumenal plug region of Sec61α that confer dominant resistance and prevent cotransin binding. These results support a model in which cotransin binding to the plug prevents lateral gating by susceptible TMDs, whereas TMDs with increased hydrophobicity and helical propensity can override the cotransin-imposed block. More generally, our characterization of a pre-integrated complex implies that dynamic interactions between the TMD and Sec61 can facilitate lateral gating and membrane integration.

## Results and discussion

### Experimental strategy

To explore the effect of cotransin on TMD integration, we used a reconstituted system comprising a mammalian translation extract supplemented with microsomal ER membranes (*Sharma et al., 2010*). When this system is programed with a truncated mRNA that lacks a stop codon, the ribosome translates to the end of the transcript and stalls, creating a synchronized population of ribosome nascent chain complexes (RNCs) of defined length that functionally engage the targeting and translocation machinery (*Gilmore et al., 1991*). We used this approach to prepare translocation intermediates of TNFα, an integral membrane protein whose single TMD mediates targeting, membrane insertion, and translocation of the C-terminal ectodomain (*Figure 1A*). Importantly, cotranslational integration of TNFα is potently inhibited by CT8, a recently described cotransin variant (*Maifeld et al., 2011*).

### CT8 blocks TMD integration into the membrane

We first established the feasibility of producing a stable complex comprised minimally of CT8, Sec61, and TNFα RNCs. Direct detection of cotransin binding to Sec61 was aided by CT7, an isosteric and equipotent analog of CT8 equipped with a diazirine for covalent photo-crosslinking to Sec61α (*MacKinnon et al., 2007*; *Figure 1—figure supplement 1A,B*). FLAG-tagged TNFα nascent chains of 126 residues (126-mers) were assembled on ER microsomes in the presence or absence of CT7, solubilized with digitonin, immunopurified with anti-FLAG beads, and analyzed for the presence of Sec61α and CT7. Sec61α specifically copurified with FLAG-TNFα RNCs, as it was not detected on anti-FLAG beads recovered from control reactions programed with HA-tagged TNFα (*Figure 1B*). Moreover, the amount of Sec61α copurifying with TNFα RNCs was identical in the absence or presence of CT7 (*Figure 1C*, lane 1 vs 2), suggesting that CT7 does not weaken the affinity of the RNC-Sec61 complex. Finally, the amount of CT7 covalently bound to Sec61α in the FLAG-TNFα immunoprecipitates was identical whether the photo-crosslinking step was carried out before or after the stringent immunopurification and washing steps (*Figure 1C*, lane 2 vs 3). Thus, CT7, Sec61, and TNFα RNCs assemble into a high-affinity ternary complex amenable to biochemical analysis.

We employed a combination of protease protection and cysteine accessibility assays to probe the local environment of the TNFα nascent chain. At the 126-mer length, TNFα RNCs should have sufficient polypeptide beyond the TMD (~75 amino acids) to have stably inserted into the Sec61 channel in the 'looped' orientation, with the TMD at least partially exposed to the lipid bilayer (*Martoglio et al., 1995*; *McCormick et al., 2003*). Consistent with this expectation, 126-mer RNCs assembled in the absence of CT8 were predominantly shielded from protease, generating a large protected fragment that consists of the TMD plus the lumenal C-terminal domain (*Figure 1D*, *Figure 1—figure supplement 1D*). This

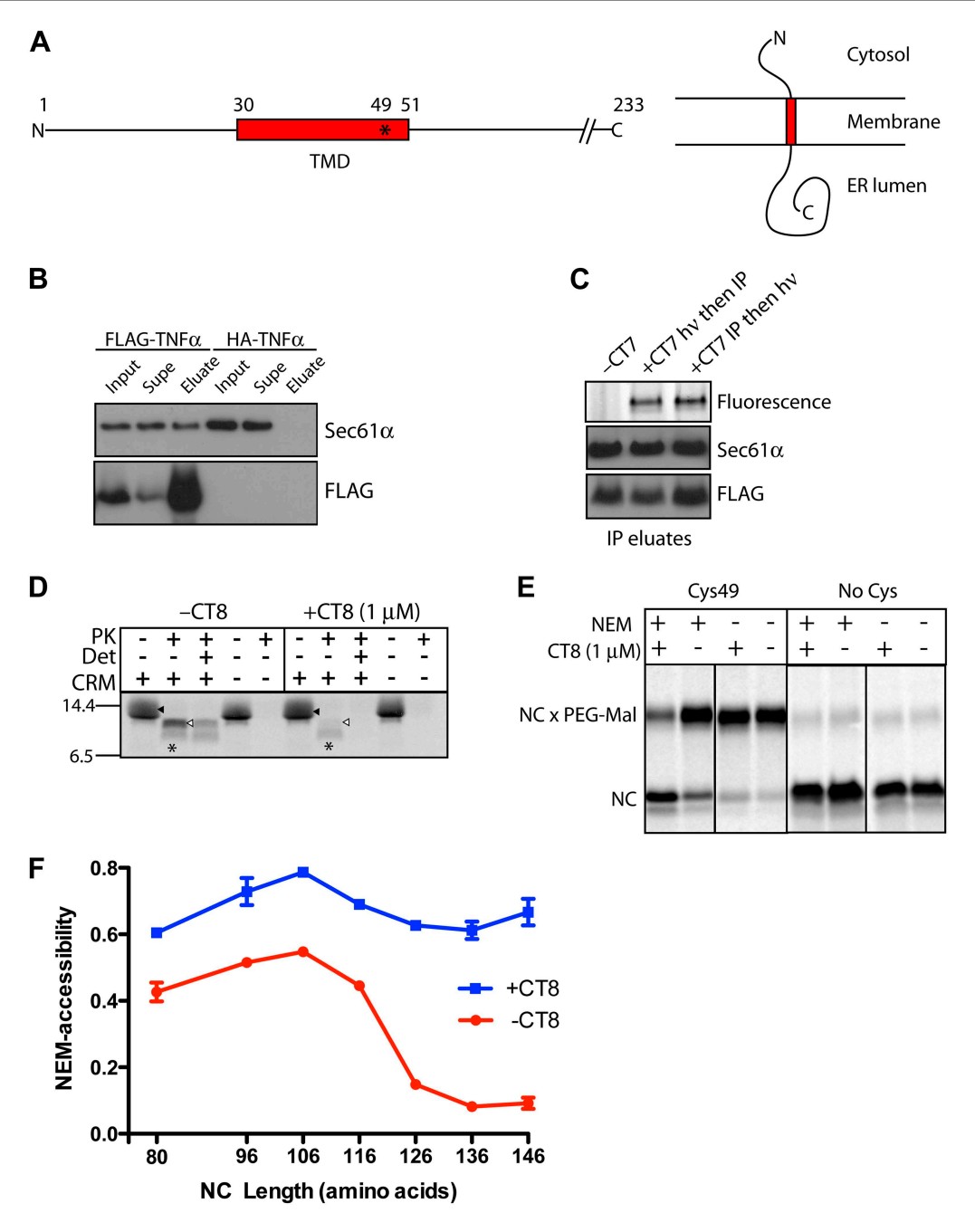

**Figure 1**. CT8 blocks TMD integration into the membrane. (**A**) Schematic of TNFα primary structure and membrane orientation. Cys49 is indicated by an asterisk. (**B**) TNFα 126-mers containing an N-terminal FLAG- or HA-tag were translated in the presence of canine rough microsomes, solubilized with 1% Deoxy BigChap (DBC), and immuno-precipitated (IP) with anti-FLAG affinity resin. IP eluates were analyzed by immunoblotting for FLAG-TNFα and Sec61α. (**C**) As in (**B**), except that microsome-targeted 126-mers were prepared in the presence or absence of the photo-affinity probe CT7 and photolyzed (*h*ν) either before (lane 2) or after immunoprecipitation (lane 3). The photo-crosslinked CT7/Sec61α adduct was detected by click chemistry with TAMRA-azide, followed by in-gel fluorescence imaging. (**D**) Protease protection assays with [35]S-labeled 126-mers translated in the presence or absence of canine rough microsomes (CRM), followed by treatment with proteinase K (PK) and TX-100 (Det) as indicated. The nascent chain and protease-protected fragment are indicated (closed and open triangle, respectively). An unidentified weak band (asterisk) was occasionally observed. (**E**) Microsome-targeted 126-mers with Cys49 or lacking

*Figure 1. Continued on next page*

*Figure 1. Continued*

all cysteines were assembled in the presence or absence of CT8 and treated with N-ethylmaleimide (NEM) as indicated. Samples were subsequently denatured with SDS, treated with PEG-maleimide (PEG-Mal), and analyzed by SDS-PAGE/phosphorimaging. NEM inaccessibility is indicated by a shift to higher molecular weight, corresponding to nascent chains (NC) modified by PEG-Mal at Cys49. (**F**) NEM accessibility of RNCs of varying length, defined as fraction of nascent chains modified by PEG-Mal, determined by phosphorimaging and normalized to a matched control reaction lacking NEM. Data represent the mean ± range of two independent experiments. For raw phosphorimaging data, see *Figure 1—figure supplement 1C*.

The following figure supplements are available for figure 1:

**Figure supplement 1**. CT8 blocks TMD integration into the membrane.

fragment was not observed in the absence of microsomes, suggesting that the TNFα TMD and ectodomain are protected by the Sec61 complex and the membrane. RNCs produced in the presence of CT8 generated sharply reduced amounts of the protected fragment, indicating that the nascent chain remained primarily in a protease-accessible compartment, despite binding tightly to the Sec61 complex (*Figure 1D*).

To probe the environment of the TMD itself, we analyzed TNFα RNCs containing a single native cysteine at position 49, near the C-terminal end of the TMD. We monitored accessibility of Cys49 to N-ethylmaleimide (NEM), which efficiently reacts with cysteine thiols exposed to an aqueous environment. In this assay, ER-targeted RNCs are first alkylated with NEM under native conditions, followed by quenching, denaturation with SDS, and modification of unreacted sulfhydryls with PEG-maleimide. Thus, PEG-modified nascent chains (identified by a molecular weight shift) represent RNCs whose Cys49 sulfhydryl was initially inaccessible to NEM. When 126-mers were assembled without CT8, Cys49 was unreactive toward NEM, whereas the same nascent chains assembled in the presence of CT8 were strongly reactive and thus protected from subsequent reaction with PEG-maleimide (*Figure 1E*). These results are consistent with the protease protection assay and demonstrate that CT8 causes 126-mers to occupy a distinct environment, one in which Cys49 in the TMD is exposed to aqueous solvent.

Applying this assay to RNCs of varying length revealed that in the absence of CT8, Cys49 progressed from a primarily NEM-accessible environment to one that was inaccessible (*Figure 1F*). This transition occurred over a narrow range of nascent chain lengths, between 116- and 126-mers. By contrast, in the presence of CT8, the TMD remained accessible to NEM at every nascent chain length tested (*Figure 1F*, *Figure 1—figure supplement 1C*). Given that efficient alkylation with NEM requires an aqueous environment, the abrupt transition observed in the absence of CT8 likely represents TMD integration into the lipid bilayer. Not only is the polypeptide sufficiently long at this stage to exit the translocon, but this is also the length at which earlier studies have observed photo-crosslinking of the TMD to phospholipids (*Martoglio et al., 1995*; *Mothes et al., 1997*; *Urbanus et al., 2001*). While the TMD may remain close to Sec61 at this stage (*McCormick et al., 2003*; *Sadlish et al., 2005*; *Devaraneni et al., 2011*; *Frauenfeld et al., 2011*; *Hou et al., 2012*), it has most likely exited the central aqueous pore of Sec61. Collectively, the above results demonstrate that CT8 prevents TMD insertion into the membrane at a step that occurs after RNC targeting to Sec61.

## CT8 stabilizes a transient pre-integrated intermediate

To gain insight into the topological relationship between the TMD and Sec61 along the integration pathway, we used bis-maleimidohexane (BMH), which can covalently crosslink solvent-accessible cysteines that lie within ~13 Å of each other. BMH crosslinking of wild-type TNFα 126-mers (native cysteines at positions 30 and 49 in the TMD) yielded no major crosslinked products (*Figure 2A*), consistent with the solvent inaccessibility of the TMD at this length. In the presence of CT8, the 126-mers crosslinked strongly to partners of ~10 and ~40 kDa (*Figure 2A*), identified by immunoprecipitation as Sec61β and Sec61α, respectively (*Figure 2—figure supplement 1*). These crosslinks were diminished when a termination codon was introduced at position 126, indicating that only tRNA-bound nascent chains crosslink to the Sec61 complex. Further supporting this interpretation, a portion of TNFα RNCs retained the tRNA during electrophoresis, and this peptidyl-tRNA was also found to efficiently crosslink to Sec61α and Sec61β (*Figure 2A*, *Figure 2—figure supplement 1B*, arrowheads). Mutagenesis of each cysteine in TNFα showed that Cys49 was uniquely responsible for the crosslinks.

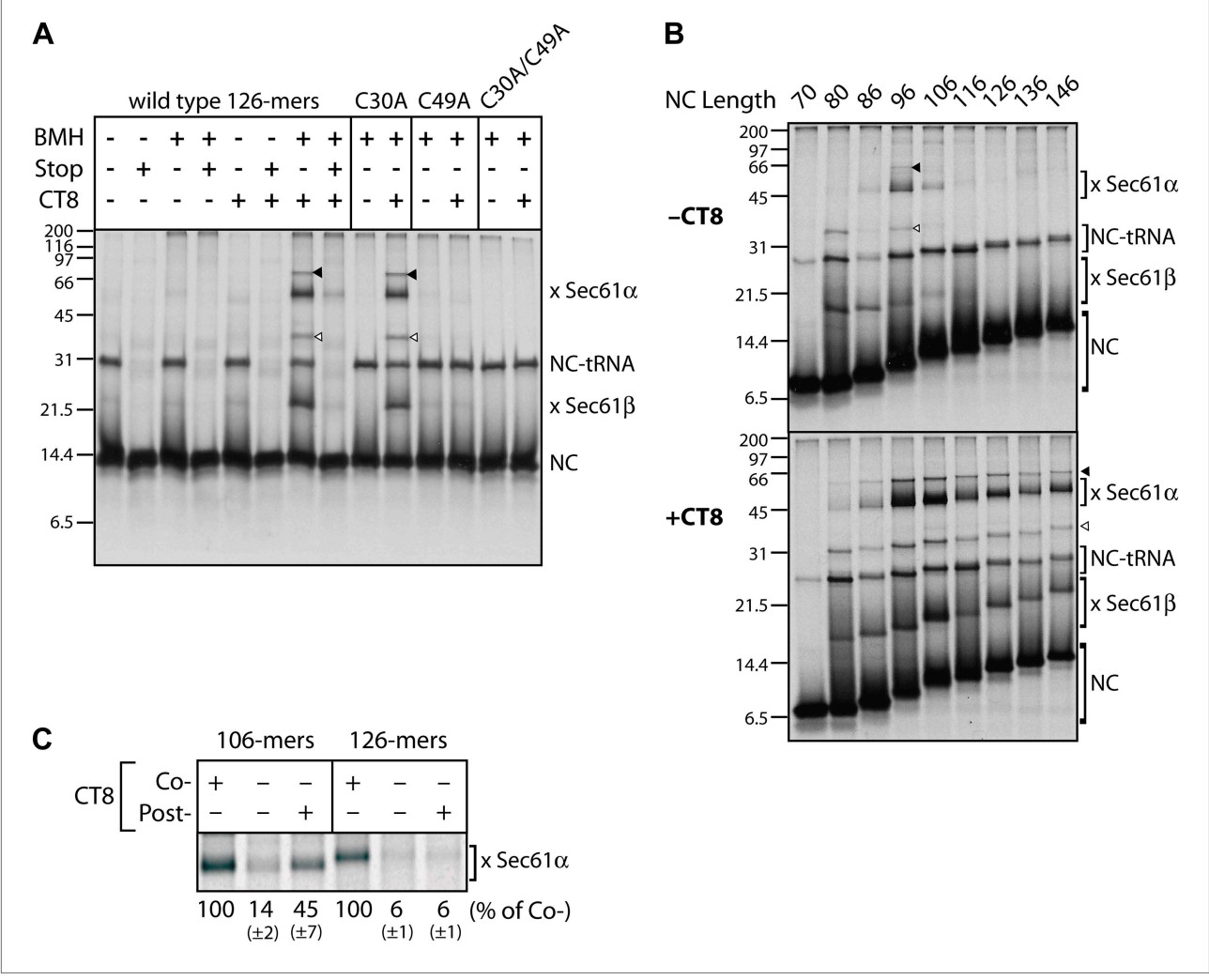

**Figure 2**. CT8 stabilizes a transient pre-integrated intermediate. (**A**) Microsome-targeted TNFα 126-mers assembled in the presence or absence of CT8 (1 μM) were treated with BMH as indicated (Stop: termination codon at position 126). Bands corresponding to the nascent chain (NC) and the NC crosslinked to Sec61α and Sec61β are indicated (for IP confirmation, see **Figure 2—figure supplement 1**). Residual NC-tRNA as well as crosslinks to Sec61α and Sec61β is also indicated (closed and open triangle, respectively). (**B**) BMH crosslinking reactions with TNFα RNCs of varying length. (**C**) Microsome-targeted TNFα RNCs of the indicated length were assembled in the continuous presence of CT8 ('Co-') or treated with CT8 post-translationally ('Post-'), followed by BMH crosslinking. NC crosslinks to Sec61α were quantified by phosphorimaging and normalized to the cotranslationally CT8-treated control. Quantified data represent the mean ± standard deviation of three independent experiments. In (**B**) and (**C**), TNFα RNCs contained a single cysteine (Cys49).

The following figure supplements are available for figure 2:

**Figure supplement 1**. CT8 stabilizes a transient pre-integrated intermediate.

We next analyzed crosslinking patterns of TNFα RNCs containing a single cysteine (Cys49 in the TMD) at various chain lengths from 70 to 146 residues, in the absence and presence of CT8. Without CT8, crosslinks first appeared between the 80-mer and Sec61β, which contains a single cysteine in its presumably unstructured cytosolic tail (**Figure 2B**). This demonstrates that the 80-mer was properly targeted to the Sec61 complex and that Cys49, predicted to have just emerged from the ribosomal exit tunnel, was exposed to the cytosol. Crosslinking to Sec61β decreased as the nascent chain was extended to 96 residues, at which point strong crosslinks to Sec61α appeared (**Figure 2B**). Given the ~15 amino acid tether between the C-terminal end of the TMD and the polypeptide exit tunnel (positions 51–65 of the 96-mer), Cys49 is likely situated within or near the cytosolic vestibule of Sec61α such that it can reach a solvent-exposed cysteine on Sec61α (via the 13 Å crosslinker). As the nascent

chain was extended beyond 96 residues, crosslinking to Sec61α decreased abruptly and was undetectable by 126 amino acids. This transition parallels the loss in NEM accessibility (*Figure 1F*) and further suggests that the TMD exits the Sec61α vestibule at this stage of nascent chain elongation. Integration of the TMD is therefore preceded by a step in which Cys49 of the TMD passes within ~13 Å of an accessible cysteine in Sec61α.

A strikingly different picture emerged when this analysis was performed in the presence of CT8. Although short nascent chains (up to 86-mers) had a similar crosslinking profile, all RNCs beyond this length were clearly distinct from the matched samples lacking CT8. Not only were the crosslinks to Sec61α and Sec61β more intense, but they also persisted beyond the point where TMD integration occurred in the absence of CT8 (*Figure 2B*). Order-of-addition experiments indicated that TMD egress from the cytosolic vestibule (operationally defined by the disappearance of TMD/Sec61α crosslinks) becomes irreversible once the nascent chain has progressed from 106- to 126-mers (*Figure 2C*). When 106-mers were first assembled in the absence of CT8, isolated by sedimentation, and then incubated with CT8 post-translationally, enhanced TMD/Sec61α crosslinks were observed. By contrast, post-translational addition of CT8 to identically prepared 126-mers had no effect, indicating that the 126-mers had crossed a kinetic point-of-no-return that could not be reversed by CT8. Taken together, these data argue: (1) CT8 stabilizes a configuration of the TMD/Sec61 complex that occurs just prior to a committed step of TMD integration, and (2) a similar pre-integrated configuration occurs in the absence of CT8 for nascent chain lengths of 96–106 amino acids. That CT8 actually enhances the initial interaction between Sec61α and the TMD (as determined by BMH crosslinking) suggests an allosteric mechanism of action.

## CT8 traps the TMD helix in a defined orientation

The CT8-arrested TMD/Sec61 complex provides a unique opportunity to characterize an otherwise transient, pre-integrated intermediate. We used BMH crosslinking to probe the environment of a cysteine residue scanned across the TMD and flanking regions of 126-mers assembled in the presence or absence of CT8 (*Figure 3A*). As expected on the basis of protease and NEM accessibility experiments (*Figure 1D,E*), 126-mers assembled in the absence of CT8 displayed no protein crosslinks to cysteines within the TMD (*Figure 3B,C*). By contrast, 126-mers prepared with CT8 showed strong crosslinks to Sec61α, with peak intensities at positions 35, 38, 42, 45, and 49 (*Figure 3B,C*). A helix-destabilizing proline mutation in the center of the TMD (V41P) abolished the periodic crosslinking pattern without significantly affecting RNC targeting to Sec61 (*Figure 3D*, *Figure 3—figure supplement 1D*). An identical periodic crosslinking pattern was observed with both shorter (96-mers) and longer (146-mers) nascent chains (*Figure 3D*, *Figure 3—figure supplement 1B,C*). The sharp periodicity of the crosslinks ($i$, $i+3$, $i+7$, etc), along with the consistent pattern observed across multiple nascent chain lengths, indicates that the TMD helix is not oriented randomly relative to Sec61α. Instead, the TMD appears to have a defined orientation, docked to Sec61α in a manner that is presumably stabilized by CT8. In the absence of CT8, a periodic crosslinking pattern across the TMD was not observed (*Figure 2B*), despite the strong crosslinks to Cys49 in the context of 96-mers (*Figure 3—figure supplement 1B*). Thus, the pre-integrated TMD/Sec61α complex likely samples multiple conformations in the absence of CT8.

Crosslinking experiments with the cysteine positioned either before or after the TMD helped define the overall topology of the nascent chain. Cysteines at positions 10 or 16 formed strong crosslinks to numerous proteins, including Sec61β (*Figure 3B*), in a pattern that was nearly identical for RNCs assembled with or without CT8. Thus, the N-terminal tail is highly accessible and localized to the cytosolic side of the membrane under both conditions.

The polypeptide segment C-terminal to the TMD of integrated 126-mers is predicted to pass through the Sec61 channel and into the ribosomal exit tunnel. In the absence of CT8, cysteines engineered along this segment in 126-mer RNCs showed strong crosslinks to Sec61α without concomitant crosslinks to Sec61β (*Figure 3B*). By contrast, the presence of CT8 caused these same cysteines to crosslink strongly to Sec61β (*Figure 3B*), again indicating that this region of the CT8-arrested nascent chain is exposed to the cytosol. We note that the N-terminal tail of CT8-arrested, pre-integrated TNFα is highly accessible to the cytosolic side of the membrane, consistent with the early establishment of a type II orientation. It is therefore unlikely that TNFα initially exposes its N-terminus to the ER lumen, as shown in a recent study of a different type II membrane protein (*Devaraneni et al., 2011*). This difference may be due to the longer, charged N-terminal tail of TNFα relative to the aquaporin-4 construct used in the previous study.

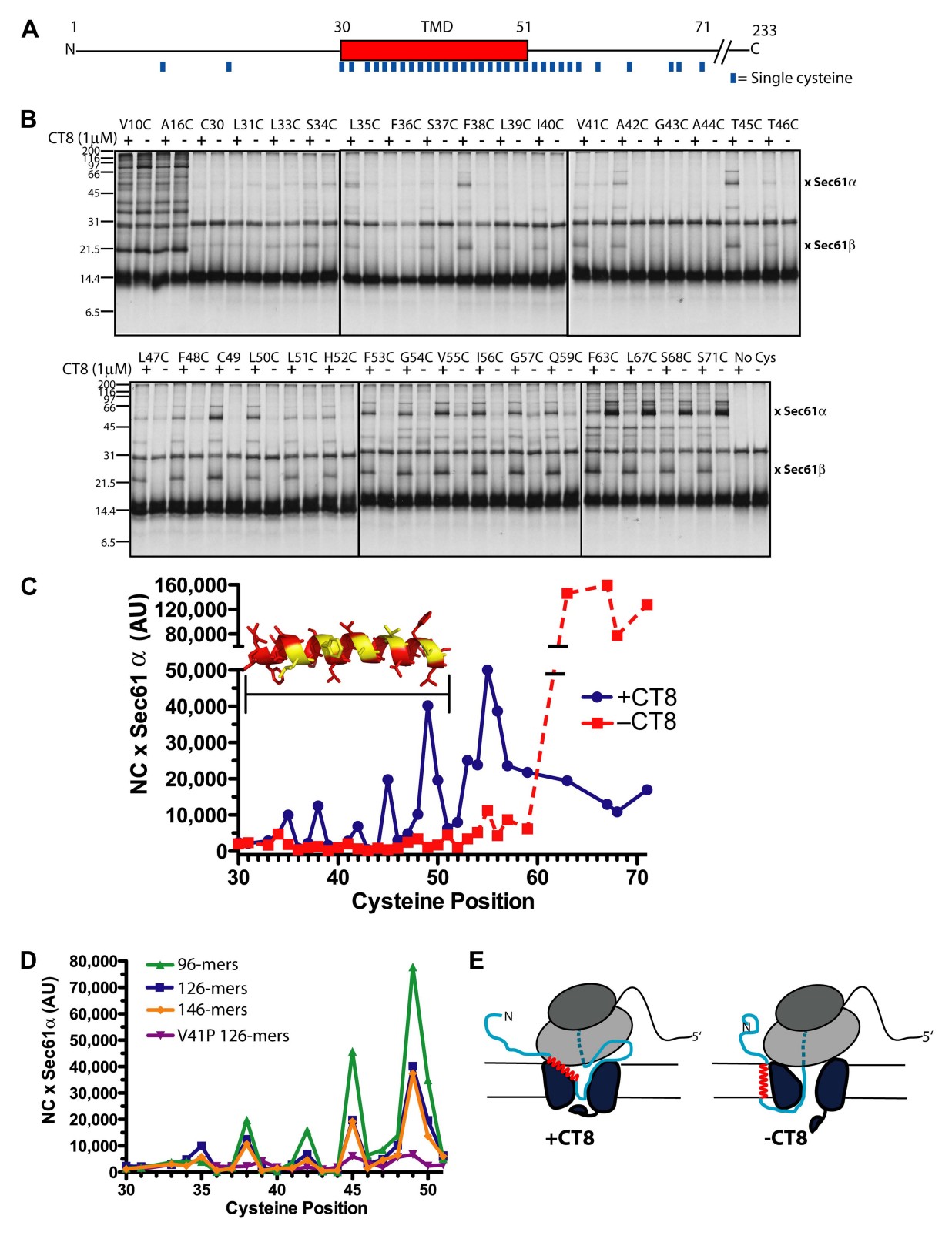

**Figure 3**. CT8 traps the TMD in a helical conformation with a defined orientation. (**A**) Positions of single cysteine mutations (blue dashes) analyzed by BMH crosslinking. (**B**) BMH crosslinking reactions of 126-mers containing a single cysteine at the indicated positions. (**C**) Crosslinking intensities (NC × Sec61α) were quantified by phosphorimaging of the gels shown in (**B**) and plotted as a function of cysteine position. Inset, TMD α-helix model (red)

*Figure 3. Continued on next page*

*Figure 3. Continued*

highlighting positions with strongest crosslinks to Sec61α (yellow). (**D**) Overlay of BMH crosslinking profiles for the indicated constructs. Crosslinking intensities were normalized to an internal standard (Cys49 126-mer) included in each experiment. For raw phosphorimaging data, see *Figure 3—figure supplement 1*. (**E**) Cartoon depicting approximate TMD disposition (red) of 126-mer assembled in the presence and absence of CT8.

The following figure supplements are available for figure 3:

**Figure supplement 1**. CT8 traps the TMD in a helical conformation with a defined orientation.

The protease protection, NEM accessibility, and BMH crosslinking data together reveal a distinct configuration of late-stage RNCs (>96-mer) assembled in the absence vs presence of CT8 (*Figure 3E*). In the absence of CT8, 126-mer RNCs are in the expected type II topology with a cytosolically disposed N-terminus and membrane-spanning TMD, followed by a C-terminal domain threading through the Sec61 channel to the ribosome. By contrast, the CT8-arrested state is distinguished by a TMD helix that has not integrated, but is instead accessible to the aqueous environment and docked to Sec61α in a specific orientation. This pre-integrated configuration is stabilized by CT8 from the 96-mer until at least the 146-mer. The degree to which the cotransin-stabilized Sec61 conformation resembles integration intermediates that form in the absence of cotransin is not clear. Cotransin may stabilize one of a continuum of conformational states adopted by Sec61 during the normal process of cotranslational integration (e.g., TMD docked to a lateral gate that is partially open toward the cytosol, but closed near the lumenal plug).

## The pre-integrated TMD docks to the cytosolic tip of the lateral gate

To identify the docking site on Sec61α, we first sought to determine which cysteine(s) in Sec61α formed BMH-induced crosslinks to the pre-integrated TMD. Unfortunately, our efforts to produce recombinant mammalian Sec61αβγ complexes on which to perform mutagenesis were unsuccessful. In considering the available structural and functional data, we reasoned that the minimal unit for TMD docking is likely to be the Sec61α/γ sub-complex. This is because the β subunit is poorly conserved across species and does not contribute substantially to the structural core of the Sec61 complex. Moreover, a stable sub-complex of mammalian Sec61α and Sec61γ was found to function in RNC targeting and translocation assays, albeit only under certain conditions (*Kalies et al., 1998*). We therefore investigated the Sec61α/γ sub-complex as an alternative system, using the criteria of stable co-association, cotransin binding, and RNC targeting as three independent indicators of correct folding and partial functionality.

Expression of FLAG-tagged Sec61α and untagged Sec61γ subunits from a single baculovirus in Sf21 insect cells produced proteins that co-purified as a stable complex with ~1:1 stoichiometry (*Figure 4—figure supplement 1A*). Importantly, the photo-affinity probe CT7 labeled FLAG-Sec61α in a manner that was competed by CT9 (*Figure 4A,B*, *Figure 4—figure supplement 1B*), a potent analog similar to CT8 (*Maifeld et al., 2011*). Moreover, when TNFα 126-mers were translated in the presence of Sf21 microsomes containing recombinant Sec61α/γ, CT8 promoted the formation of BMH crosslinks between Cys49 and recombinant Sec61α (*Figure 4C*, *Figure 4—figure supplement 1C*). In each of these experiments, recombinant FLAG-Sec61α behaved similarly to native Sec61α in canine pancreatic microsomes, suggesting that the recombinant complex is correctly folded and sufficiently functional to analyze its interaction with the TNFα 126-mer.

To determine which of the eight native cysteines in Sec61α crosslinked to the TMD, each was mutated to alanine or serine. All Sec61α mutants co-expressed with Sec61γ at levels similar to the wild type and were properly folded, as assessed by specific photo-affinity labeling with CT7 (*Figure 4B*, *Figure 4—figure supplement 1D*). A single mutation in Sec61α (C13A) abolished BMH crosslinking to the TMD, whereas each of the other cysteine mutants crosslinked efficiently in the presence of CT8 (*Figure 4C*). Based on a homology model of the human Sec61 complex derived from the crystal structure of an archaeal ortholog (*Van den Berg et al., 2004*; *Erdmann et al., 2009*), Cys13 is predicted to be in a short α-helix that lies parallel to the cytosolic face of the membrane, just outside the cytosolic vestibule and adjacent to TM3 (*Figure 4E*).

To refine the location of the pre-integrated TMD, non-conserved residues throughout the cytosolic vestibule of Sec61α were individually mutated to cysteine in the background of the C13A mutation (a mutant Sec61α construct lacking all 8 native cysteines was non-functional in the CT7 photo-crosslinking assay). Like the single mutants, all of the double mutants co-expressed with Sec61γ at similar

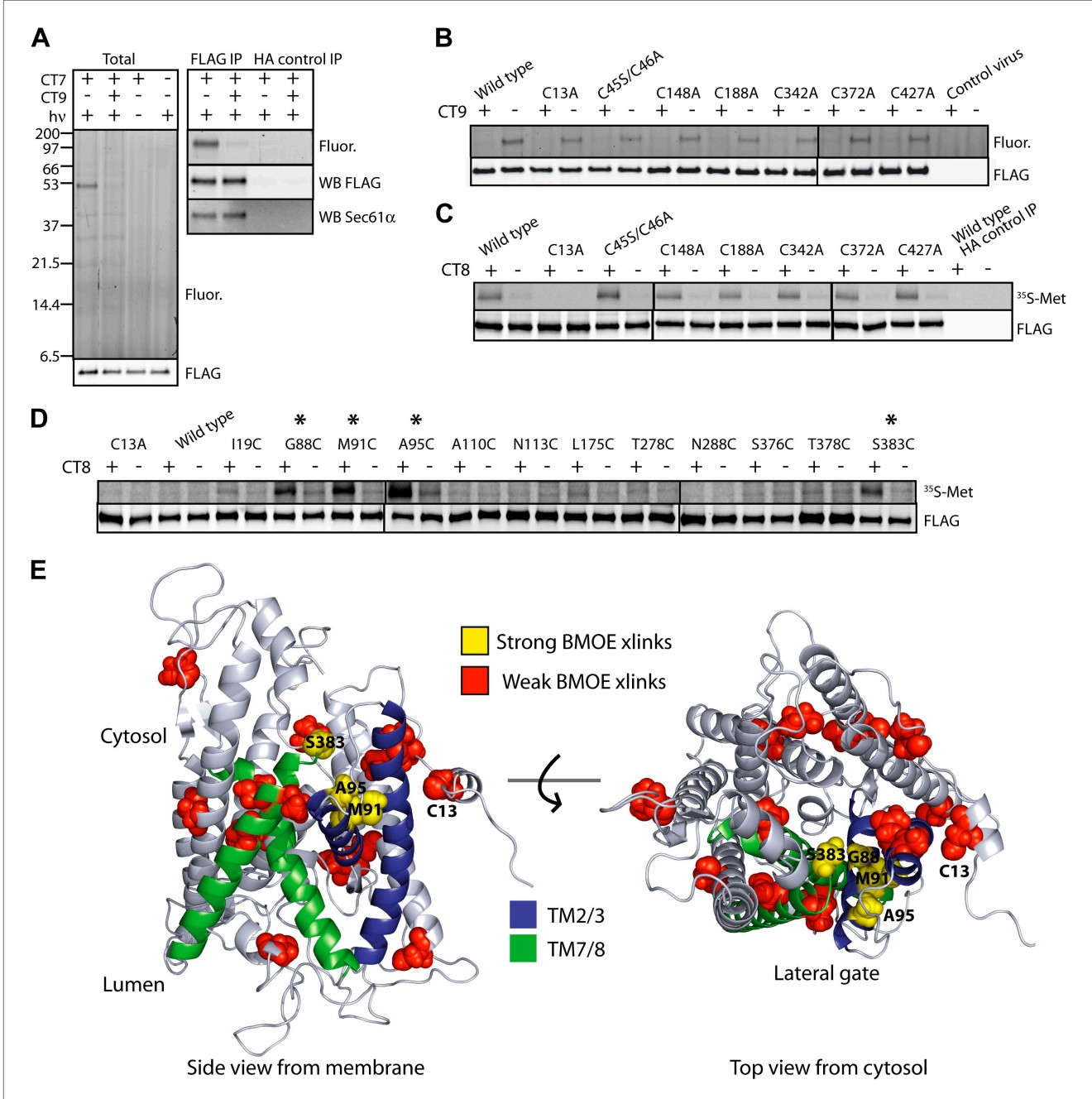

**Figure 4**. The pre-integrated TMD docks to the cytosolic tip of the lateral gate. (**A**) Cotransin binding activity of recombinant FLAG-Sec61α. Microsomes from Sf21 cells expressing FLAG-Sec61α and Sec61γ were incubated with CT7 (50 nM) in the presence or absence of excess CT9 (10 μM). Samples were photolyzed (hv), and analyzed as in *Figure 1C* (left panel). Samples were immunoprecipitated under denaturing conditions with anti-FLAG or anti-HA (control) beads and analyzed as previously (right panel). (**B**) Photo-affinity labeling with CT7 as in (**A**) with microsomes from FLAG-Sec61α/γ mutant or control baculovirus-infected Sf21 cells. (**C**) 35S-labeled TNFα 126-mers (Cys49) were translated in the presence of FLAG-Sec61α/γ microsomes and subjected to BMH crosslinking. After immunoprecipitation with anti-FLAG beads, eluates were analyzed by autoradiography and immunoblotting. (**D**) As in (**C**), except that crosslinking reactions were performed with BMOE instead of BMH. See *Figure 4—figure supplement 1* for further characterization of recombinant FLAG-Sec61α/γ. (**E**) Homology model of human Sec61α (***Erdmann et al., 2009***) (based on PDB entry:1RH5), highlighting positions of native and engineered cysteines (yellow = strong, red = weak crosslinks to TNFα TMD Cys49). Lateral gate helices are shown in blue (TM2b/3) and green (TM7/8).

The following figure supplements are available for figure 4:

**Figure supplement 1**. The pre-integrated TMD docks to the cytosolic tip of the lateral gate.

levels and specifically bound CT7 (*Figure 4—figure supplement 1E*). To enforce a more stringent distance constraint in the cysteine crosslinking experiments, we used bis-maleimidoethane (BMOE), the shortest available bis-maleimide crosslinker (~8 Å). Sec61α cysteine-substitution mutants were then analyzed for crosslinking to Cys49 in TNFα 126-mers.

Unlike the longer BMH crosslinker (~13 Å), BMOE failed to promote crosslinking of the TMD to Cys13 of wild-type Sec61α (*Figure 4D*), suggesting that BMOE is a more stringent reporter of Cys–Cys proximity. Similarly, crosslinks to most of the engineered cysteines within the cytosolic vestibule of Sec61α were weak or undetectable (*Figure 4D*). Notable exceptions were cysteine substitutions at Gly88, Met91, Ala95, and Ser383, each of which showed strong CT8-enhanced crosslinks. Mapping the location of these engineered cysteines onto a model of human Sec61α revealed a clear hotspot where the cytosolic face of TM2b meets the cytosolic tip of TM8 (*Figure 4E*). This region of Sec61α represents the cytosol-exposed tip of the lateral gate, shown here to lie within ~8 Å of the TMD (Cys49) in the CT8-stabilized, pre-integrated configuration.

## Resistant and sensitive TMDs pass through a common pre-integrated intermediate

Previous studies have shown that the cotransin sensitivity of a given secretory protein can be altered by mutations in the signal sequence (*Harant et al., 2006*, *2007*). However, a mechanistic explanation for these effects has remained elusive. Given the intimate association between the TMD and the cytosolic face of TM2b and TM8, we reasoned that certain TMD mutants should be able to overcome the CT8-imposed barrier to integration via productive interactions with the lateral gate helices and/or membrane lipids. Comparison of sensitive and resistant TMDs at multiple points along the integration pathway (i.e., at multiple nascent chain lengths) should allow us to: (1) test whether they pass through a common intermediate and (2) define the point at which their fates diverge. Finally, analysis of diverse TMD mutants should provide insight into the biophysical basis of CT8 sensitivity and resistance.

We identified TNFα TMD mutants with a broad range of CT8 sensitivity (see below). We first present a detailed analysis of the double mutant, T45L/T46L. Inhibiting cotranslational integration of this mutant required 13-fold higher concentrations of CT8 as compared to wild-type TNFα (IC$_{50}$ of 1040 nM and 80 nM, respectively). BMH crosslinking of successively longer T45L/T46L RNCs assembled without CT8 showed a pattern that was similar to wild-type TNFα: crosslinks to Sec61α were observed over a narrow length range, 86- to 96-mers, beyond which the crosslinks abruptly disappeared (*Figure 5A*). As with the wild type, low nanomolar concentrations of CT8 led to increased crosslinking between the T45L/T46L mutant and Sec61α, but this was true only at short nascent chain lengths (*Figure 5A,B*). At longer chain lengths, crosslinking of the mutant TMD to Sec61α was not observed, consistent with its successful integration into the membrane beyond the 96-mer stage (*Figure 5A,C*). A cysteine scan across the TMD of wild-type and mutant 96-mers revealed identical crosslinking patterns (*Figure 5D*, *Figure 5—figure supplement 1*), suggesting that the T45L/T46L TMD docks to Sec61α in a specific conformation and orientation similar to the wild type. Unlike the wild type, the mutant TMD escapes from the CT8-mediated arrest point and integrates into the membrane as the nascent polypeptide continues to elongate.

Given that CT8 arrests wild-type and T45L/T46L 96-mers in the same pre-integrated conformation, the question arises as to how the mutant TMD is able to escape and pass through the lateral gate. One potential explanation is the increased overall hydrophobicity of the T45L/T46L mutant, which is predicted to have a greater thermodynamic driving force for membrane integration (*Hessa et al., 2007*; *Ojemalm et al., 2011*). While mutating multiple polar residues to leucine resulted in increased CT8 resistance (*Figure 6B*), further mutational analysis revealed that overall hydrophobicity is not the sole determinant. Hydrophobic to lysine mutations in the N-terminal region of the TMD, for example, led to increased CT8 resistance, whereas such mutations in the central or C-terminal regions had the opposite effect (*Figure 6C*). This may be mechanistically related to the observation that positively charged flanking residues influence TMD orientation ('positive-inside rule') and facilitate membrane integration (*Hessa et al., 2007*; *Lerch-Bader et al., 2008*), potentially due to interactions with negative charges on or near the cytoplasmic face of Sec61 (*Goder et al., 2004*; *Frauenfeld et al., 2011*). Proline-scanning mutagenesis across the TMD resulted in similar position-dependent effects, conferring increased CT8 resistance when introduced into the N-terminal region and increased sensitivity in the central region (*Figure 6D*).

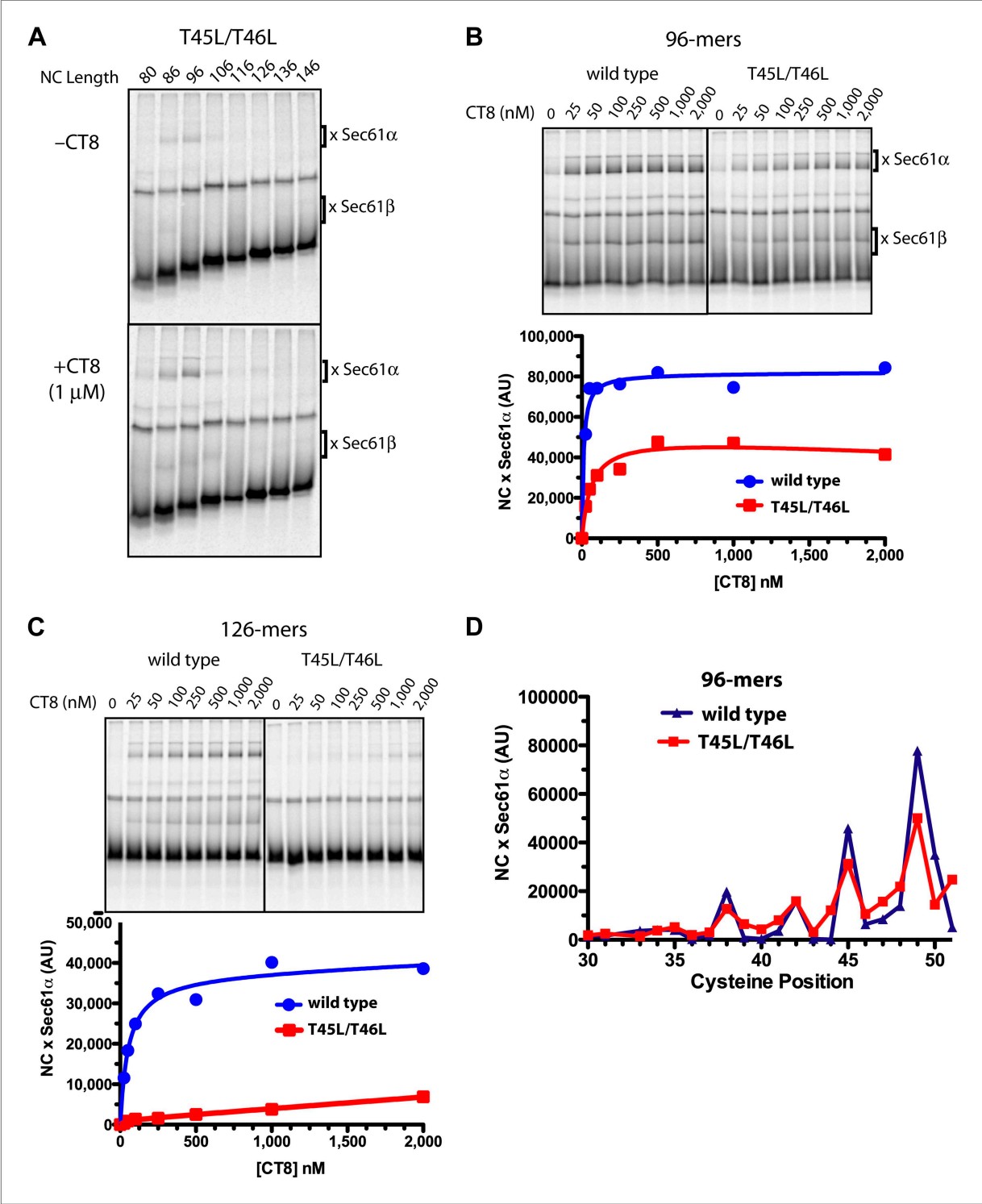

**Figure 5**. Resistant and sensitive TMDs pass through a common pre-integrated intermediate. (**A**) BMH crosslinking reactions with T45L/T46L TNFα RNCs of varying length, assembled in the presence or absence of CT8. (**B**) BMH crosslinking reactions of wild-type or T45L/T46L 96-mers in the presence of increasing concentrations of CT8. Crosslink intensities (× Sec61α) were quantified by phosphorimaging and plotted as a function of [CT8]. (**C**) As in (**B**), except 126-mers were used. (**D**) T45L/T46L 96-mers containing single cysteines at the indicated positions were assembled in the presence of CT8 (1 μM) and subjected to BMH crosslinking. Crosslink intensities (× Sec61α) were quantified by phosphorimaging (see *Figure 5—figure supplement 1* for raw data) and plotted as a function of cysteine position. Data from 'wild-type' 96-mer cysteine scan (*Figure 3D*) are shown for comparison.

*Figure 5. Continued on next page*

*Figure 5. Continued*

The following figure supplements are available for figure 5:

**Figure supplement 1**. Resistant and sensitive TMDs pass through a common pre-integrated intermediate.

Collectively, the mutagenesis data suggest that helical propensity, hydrophobicity, and charge distribution can influence TMD movement through the lateral gate in opposition to CT8 binding. This apparent competitive relationship is indicated by shifts in the CT8 dose-response curves as a result of TMD mutations (*Figure 6*). Rather than attribute these differences solely to thermodynamic partitioning of the TMD into the lipid phase (*Hessa et al., 2005*, *2007*), we propose that escape from the CT8 arrest point involves dynamic interactions between the TMD and the hydrophobic helices of the lateral gate (TM2/3 and TM7/8), in addition to interactions with membrane lipids. We speculate that these combined interactions, coupled with nascent chain elongation, ultimately cause the lateral gate to open, a conformational change that is opposed by cotransin binding to Sec61.

## Mutations in the lumenal plug of Sec61α confer resistance to cotransins

To gain further molecular insight into the mechanism of CT8-mediated arrest, we sought to determine its binding site on Sec61α. Repeated attempts to identify the CT7 photo-crosslinking site by mass spectrometry were unsuccessful, likely due to the challenge of analyzing heterogeneous, hydrophobic peptides derived from a CT7/Sec61α photo-conjugate. Given that CT8 and its more potent analog CT9 are cytotoxic to certain cancer cell lines (*Figure 7A* and our unpublished results), we reasoned that it should be possible to identify resistance-conferring alleles of Sec61α by genetic selection. With few exceptions, dominant resistance mutations that reduce a drug's binding affinity localize to the drug's binding site or its immediate vicinity; confidence in this interpretation is especially high when multiple resistance mutations localize to the same site (*Campbell et al., 2001*; *Shah et al., 2002*; *Wacker et al., 2012*), although remote allosteric effects on drug binding are also possible.

We exposed the DNA repair-defective tumor cell line HCT-116 (*Wacker et al., 2012*) to cytotoxic concentrations of CT9, our most potent cotransin variant (see 'Materials and methods' for details). After 9–12 days, during which time most cells died, several colonies were isolated and found to display varying degrees of resistance to both CT8 and CT9 (*Figure 7A*, *Figure 6—figure supplement 1A*, respectively). After amplifying the *S61A1* coding sequence from total RNA, Sanger sequencing revealed that 11 of 11 resistant cell lines had one of five single-nucleotide transitions (all heterozygous) at four amino acid positions (*Figure 7A*, *Figure 6—figure supplement 1B*). All five mutations associated with CT8 resistance cluster in the same region of Sec61α (*Figure 7D*), at the interface between the plug (R66I, R66G, G80V, S82P) and the C-terminal end of TM3 (M136T). This interface defines the side of the lateral gate that is closest to the ER lumen. The fact that five independent resistance mutations localize within ~10 Å of each other to the lumenal plug region argues that this is the cotransin binding site.

We characterized two mutations in greater detail, one in the plug (R66I) and the other at the lumenal end of TM3 (M136T). To determine whether these mutants support TNFα integration in the presence of CT8, we generated stable cell lines that express wild-type or mutant Sec61α constructs from a tetracycline-inducible promoter. For these experiments, we used HEK293 cells, whose viability is unaffected by CT8 treatment for at least 72 hr. Upon induction with tetracycline, both wild-type and mutant Sec61α transgenes (untagged) were expressed at similar levels as the endogenous protein (*Figure 6—figure supplement 1C*). Whereas CT8 potently inhibited TNFα expression in cells with the wild-type Sec61α transgene (IC$_{50}$ ~ 50 nM), it had little effect in cells carrying either the M136T or R66I mutant (*Figure 7B*). These results indicate that the Sec61α mutants assemble into functional translocons and that the M136T and R66I mutations are sufficient to confer dominant resistance to CT8. Because we could not easily distinguish endogenous Sec61α from the mutants expressed in HEK293 cells, we measured cotransin binding to recombinant Sec61α/γ overexpressed in Sf21 insect cells, as described in *Figure 4*. CT7 photo-crosslinking assays revealed specific binding to wild-type Sec61α, but greatly reduced and undetectable binding to the M136T and R66I mutants, respectively (*Figure 7C*). Although the mutations may have subtle effects on Sec61 function, the CT7 photo-crosslinking data argue that reduced cotransin binding causes resistance in the cell proliferation and TNFα expression assays.

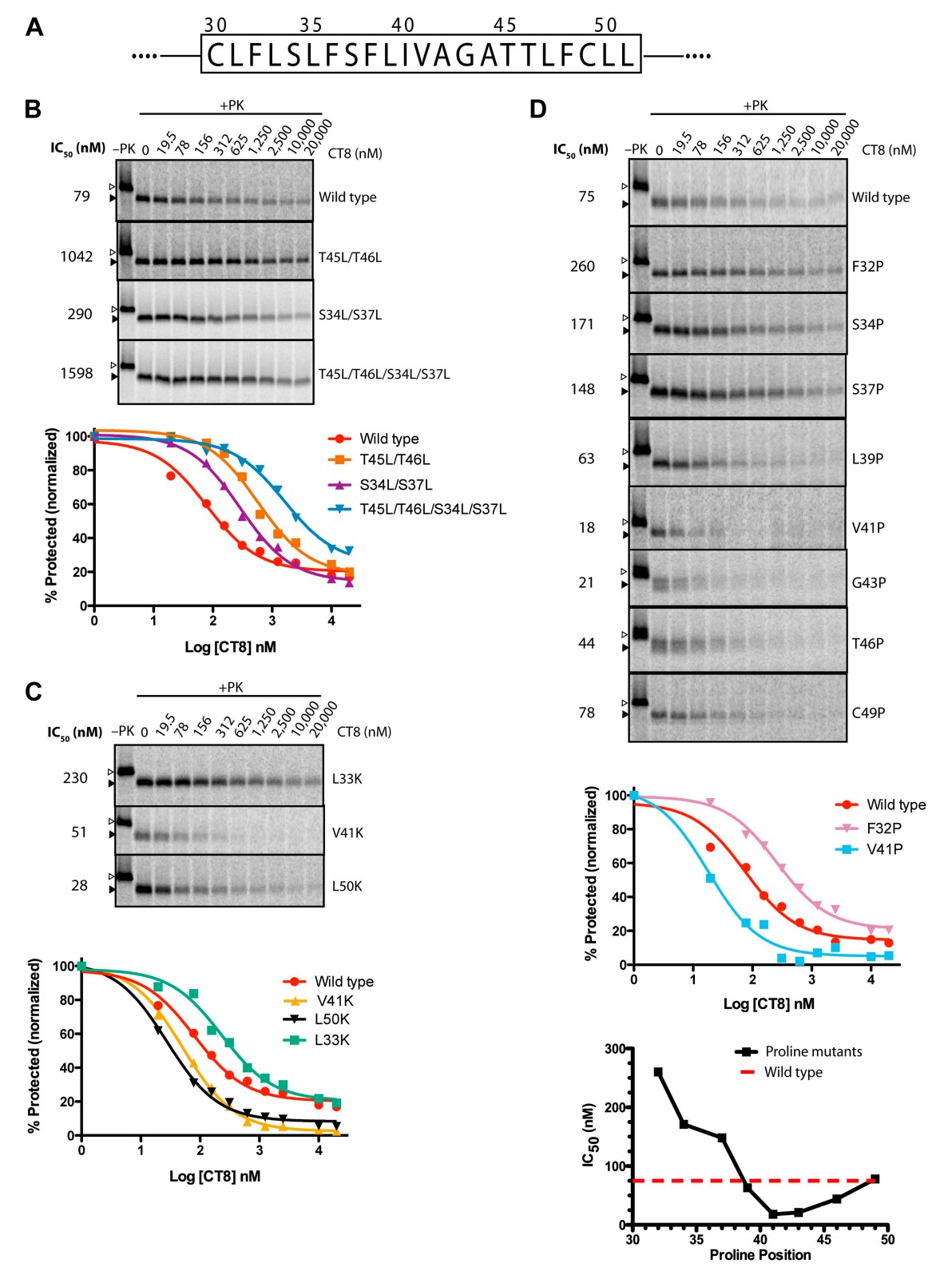

**Figure 6**. TMD sequence and biophysical properties determine CT8 sensitivity. (**A**) Amino acid sequence of the TNFα TMD. (**B**) Protease protection assays (proteinase K, PK) of full-length TNFα polar-to-leucine mutants as a function of increasing CT8 concentration. Translation reactions were carried out in the presence of increasing concentrations of CT8 or DMSO control, followed by digestion with PK as previously described (**Sharma et al., 2010**). Full-length
*Figure 6. Continued on next page*

*Figure 6. Continued*

TNFα and the protease-resistant fragment are indicated by open and closed triangles, respectively. (**C**) As in (**B**), except with hydrophobic-to-lysine mutants. (**D**) As in (**B**), except with proline-scanning mutagenesis. $IC_{50}$ values were plotted as a function of the position of the corresponding proline mutation. To compute $IC_{50}$ values, protease-resistant fragments were quantified by phosphorimaging, normalized to the DMSO control, and fitted to a three-parameter equation using GraphPad Prism software. Representative curve fits in each set (**B**–**D**) are shown. Based on the online TMD prediction algorithm (http://dgpred.cbr.su.se/), no mutation was expected to change the N-terminal or C-terminal boundaries of the TMD.

The following figure supplements are available for figure 6:

**Figure supplement 1**. Mutations in the plug/lateral gate interface confer resistance to cotransin.

## Conclusions and perspective

Structural, mutagenesis, and crosslinking analyses have all converged on the lateral gate as the site where hydrophobic segments exit the central pore of Sec61 and enter the lipid bilayer (***du Plessis et al., 2009***; ***Egea and Stroud, 2010***; ***Frauenfeld et al., 2011***; ***Plath et al., 1998***; ***Trueman et al., 2011***; ***Tsukazaki et al., 2008***; ***Zimmer et al., 2008***). However, the mechanism and timing of TMD egress, along with the role of the TMD itself in the integration process, have remained unclear. In this study, we have exploited a small-molecule inhibitor of cotranslational integration (cotransin, CT8) to trap and interrogate a nascent TMD prior to its exit from the cytosolic vestibule. By analyzing recombinant cysteine mutants of Sec61α, we identified a TMD docking site near the cytosolic tip of the lateral gate. This intimate association suggests that the TMD helix may facilitate opening of the lateral gate. Indeed, such a gating transition may underlie the recently described 'pulling force' exerted by the translocon on a nascent TMD just before its integration into the membrane (***Ismail et al., 2012***).

*Figure 8* depicts a model that places our biochemical data in the context of Sec61/SecY structures determined by x-ray crystallography and cryoelectron microscopy. In this model, RNC targeting to Sec61 allows partial opening of the lateral gate toward the cytosol, as observed in a crystal structure of SecYE bound to a Fab fragment (***Tsukazaki et al., 2008***). In the SecYE/Fab structure, separation of TM2b from the cytosolic end of TM8 creates a notch in the lateral gate, which we propose to be the initial docking site for a nascent TMD after its release from SRP (*Figure 8*, middle). At the 96-mer stage, docking of the TMD to this site enables BMH crosslinking to Sec61α (*Figure 2B*). As the nascent chain elongates, interhelical contacts that seal the lateral gate are progressively destabilized. This key transition, which is opposed by CT8 binding (most likely to the plug), leads to complete intercalation of the TMD between helices TM2/3 and TM7/8 of the lateral gate, concomitant with exposure of the TMD to membrane lipids (*Figure 8*, right). As indicated by the pronounced rightward shifts in the CT8 dose-response curves (*Figure 6*), TMDs with greater hydrophobicity and helical propensity are better able to progress to this state, presumably in kinetic competition with CT8.

Our discovery of five resistance mutations in Sec61α that localize to the lumenal plug region suggests that CT8 binds proximal to this site. This conclusion is further strengthened by the observation that R66I and M136T mutations abrogate CT7 binding (*Figure 7C*), although we cannot exclude the possibility that mutations in the plug perturb cotransin binding to a remote site. Given that binding to the lumenal plug arrests the TMD in its cytosolic docking site >20 Å away, cotransins appear to exhibit characteristics of both allosteric and competitive inhibitors. We note that the closed conformation of the channel (***Van den Berg et al., 2004***) cannot accommodate a molecule the size of CT8, whereas a cavity, outlined by the identified mutations and roughly large enough for CT8, is apparent in structures with a partially open lateral gate (***Egea and Stroud, 2010***). We speculate that CT8 binding stabilizes interactions between the plug, TM3, and TM7, thereby sealing the lateral gate at its lumenal end in a manner similar to that proposed for the Q129L and N302L mutations in yeast Sec61p (***Trueman et al., 2012***). In this view, CT8 acts as a molecular wedge that impedes TMD integration by blocking displacement of the plug. Alternatively, CT8 binding may itself result in the displacement or structural rearrangement of the plug; here again, CT8 would act as a molecular wedge that prevents TMD movement through the lumenal end of the lateral gate. This model is suggested by a recent electron crystallographic structure of the SecYEG complex in which a signal peptide bound to the lipid-exposed face of the lateral gate displaces the plug 10 Å relative to apo-SecYEG (***Hizlan et al., 2012***). Further structural studies are needed to provide an atomic-resolution view of how cotransins engage Sec61 to arrest TMD integration.

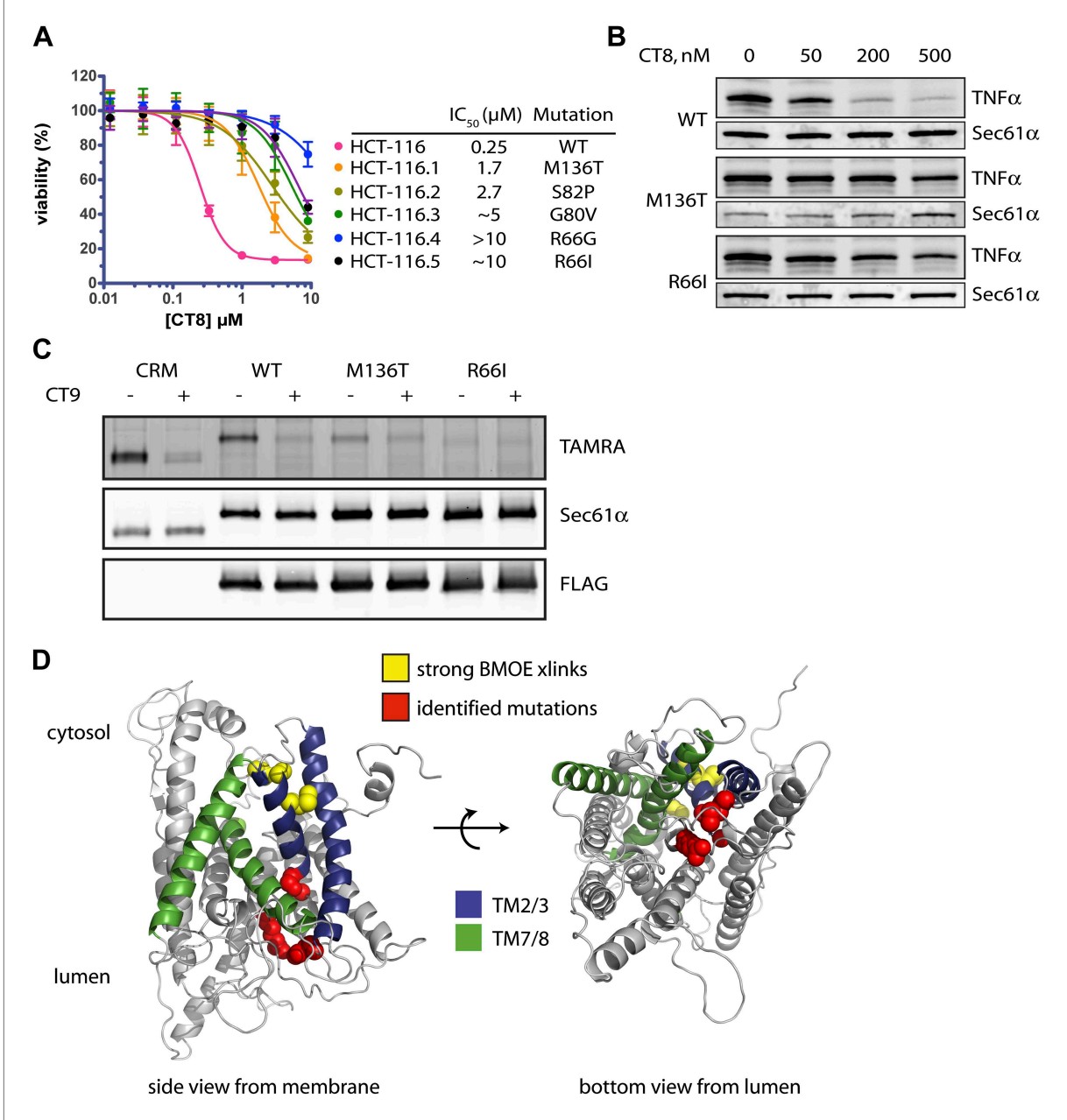

**Figure 7**. Mutations in the lumenal plug of Sec61α confer resistance to cotransins. (**A**) Parental HCT-116 cells and resistant clones were treated with increasing concentrations of CT8 for 72 hr, and viability was assessed by the Alamar Blue assay (mean ± S.D., n = 4). (**B**) HEK293-FRT cells stably expressing wild-type or mutant Sec61α were transfected with a plasmid encoding TNFα and treated with increasing concentrations of CT8. After 24 hr, TNFα expression was analyzed by immunoblotting. (**C**) CT7 photo-crosslinking to recombinant wild-type and mutant FLAG-Sec61α. Microsomes were incubated with 250 nM CT7 in the presence or absence of excess CT9 (10 μM), photolyzed, and analyzed by in-gel fluorescence and immunoblotting as in *Figure 1C*. (**D**) Homology model of human Sec61α showing the location of cotransin resistance mutations (red) and the TMD docking site (yellow). Lateral gate helices colored as in *Figure 4E*.

# Materials and methods

## Antibodies, proteins, and reagents

The following commercially available primary antibodies and antibody resins were used: anti-FLAG M2 antibody and affinity matrix (Sigma, St. Louis, MO), anti-HA affinity matrix (Roche, Basel, Switzerland), anti-Sec61γ (Proteintech Group, Chicago, IL), anti-Sec61α (Novus Biologics, Littleton, CO), anti-human TNFα (R&D Systems MAB2101, Minneapolis, MN). The antibody directed against Sec61β was previously

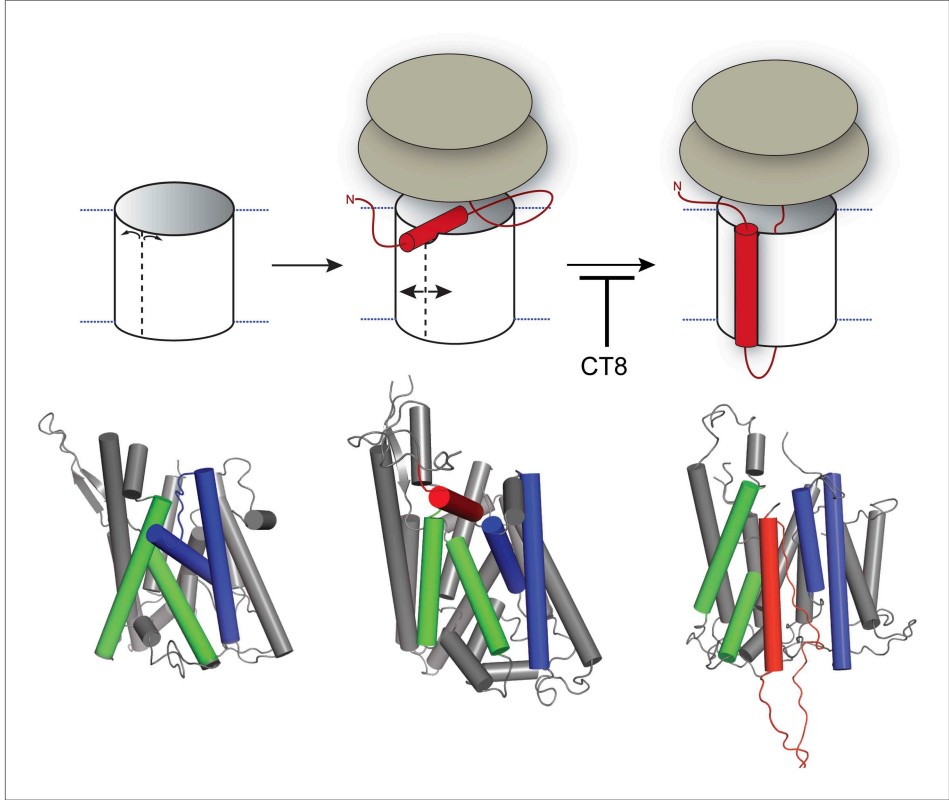

**Figure 8**. Model for cotransin-mediated inhibition of TMD integration. Following RNC targeting to Sec61 (left panel), the TMD docks to a notch in the cytosolic tip of the lateral gate created by the separation of TM2b from TM8 (middle panel). This otherwise transient, pre-integrated configuration is stabilized by CT8 binding to the lumenal plug region of Sec61α. Interactions between the TMD and the lateral gate facilitate gate opening and TMD movement to a second site on the lipid-exposed face of the lateral gate (right panel). CT8 blocks TMD movement from the cytosolic docking site to the external, lipid-exposed site. Structural models (bottom) were adapted from PDB entries 1RH5 (left panel, closed lateral gate), 2ZJS (middle panel, pre-integrated configuration), and 3J00/3J01 (right panel, integrated configuration). The model for the pre-integrated TMD configuration was approximated by manually placing the TMD (modeled as an α-helix) into the cytosolic vestibule between TM2b and TM7 (PDB entry: 2ZJS), consistent with the BMOE crosslinking data. Lateral gate helices TM2b/3 and TM7/8 are blue and green, respectively, while the TMD helix is red.

described (***Fons et al., 2003***). Secondary antibodies conjugated to HRP (Santa Cruz Biotechnology, Dallas, TX) or infrared-dyes (LI-COR Biosciences, Lincoln, NE) were purchased. Preparations of rabbit reticulocyte lysate (***Sharma et al., 2010***) and canine rough microsomes (***Walter and Blobel, 1983***) have been previously described. CT7 and CT8 were prepared as previously described (***MacKinnon et al., 2007***). CT9 was prepared as previously described (***Maifeld et al., 2011***). The sources of other reagents are noted in the text and were used without further purification.

## Plasmid and DNA construction

Site-directed mutagenesis was performed using the QuikChange method (Stratagene, La Jolla, CA) and all constructs were verified by DNA sequencing. The plasmid encoding human TNFα was previously described (***Maifeld et al., 2011***). DNA templates encoding truncated TNFα constructs were prepared by PCR using a forward primer that contained a T7 promoter (bold) and a Kozak consensus sequence (underlined) followed by a region complementary to the 5'-end of TNFα (5'-**GCCTAATACGACTCACTATAGG** GAGACCATGAGCACTGAAAGCATGATCCGG-3'). The reverse primer annealed to various regions of the TNFα coding region and introduced three additional methionines at the C-terminus to improve detection of translated products by autoradiography. The indicated length of the nascent chain includes the three terminal methionines. A similar PCR-based strategy was used for production of full-length TNFα DNA templates except the reverse primer included a stop codon. DNA templates encoding

N-terminally tagged TNFα constructs were prepared by PCR using forward primers encoding either the 3x-FLAG or 3x-HA epitopes immediately upstream of the TNFα translational start site.

Construction of the composite MultiBac baculovirus expression vector followed the previously described strategic guidelines (*Fitzgerald et al., 2006*). Briefly, human Sec61α and Sec61γ genes were amplified by PCR using templates kindly provided by Professor Tom Rapoport (Harvard Medical School), and a 3x-FLAG tag was introduced into the N-terminus of Sec61α. The Sec61α subunit was then cloned into the MCS-1 of vector pUCDM by In-Fusion cloning according to manufacturer's instructions (Clontech, Mountain View, CA). The Sec61γ subunit was cloned into SacI/HindIII sites (MCS-2) of vector pKL. The two vectors were then fused by Cre-recombination yielding a 'master plasmid', which was used to transform DH10α MultiBac cells for preparation of the MultiBac expression vector. Site-directed mutagenesis to create Sec61α mutants was performed on the master plasmid. All constructs were verified by DNA sequencing.

## SDS-PAGE, autoradiography and western blotting

Unless otherwise noted, SDS-PAGE was performed with 12% Tris/Tricine polyacrylamide gels. Prior to auto-radiography, gels were stained with Coomassie brilliant blue to confirm equal protein loading and dried under vacuum. For quantitative autoradiography, dried gels were exposed to a storage phosphorous screen (GE Healthcare, San Francisco, CA), imaged on a Typhoon 9400 phosphorimager (Amersham), and the images were quantified using ImageJ software (NIH). Dose-response data were normalized to the DMSO control and fitted with a three-parameter equation using GraphPad Prism (GraphPad Software). For qualitative autoradiography, dried gels were exposed to Biomax MR film (Kodak , Rochester, NY).

For western blotting, proteins were transferred to nitrocellulose membranes. Following blocking of the membranes with 5% milk in TBST (50 mM Tris, 150 mM NaCl, 0.05% TX-100, pH 7.6), membranes were incubated with the appropriate primary antibodies at the following dilutions: 1:10,0000 (anti-Sec61α), 1:20,000 (anti-Sec61β), 1:500 (anti-Sec61γ), 1:20,000 (anti-FLAG). Blotting for Se61γ typically required overnight incubation at 4°C with the primary antibody. Following incubation with primary antibodies, membranes were washed and incubated with the appropriate HRP-conjugated secondary antibodies followed by chemiluminescent detection (GE Healthcare). Alternatively, blots were incubated with the appropriate infrared dye-conjugated secondary antibodies followed by imaging on an Odyssey infrared fluorescent scanner (LI-COR Biosciences).

## Cell-free translation/translocation assays

Cell-free transcription, translation, translocation, and Protease K (PK) protection assays were performed as previously described (*Sharma et al., 2010*). Briefly, DNA templates encoding full-length or truncated TNFα constructs were transcribed with T7 Polymerase (New England Biolabs, Ipswich, MA) for 1 hr at 37°C and used immediately in the subsequent translation/translocation reactions. Translocation reactions were assembled at 0°C in the presence of CTs (added from a 100 × stock in DMSO) or an equivalent volume of DMSO. Unless otherwise indicated, reactions included $^{35}$S-Methionine (Perkin Elmer, Waltham, MA, 2 μCi per 10 μl translation), and either canine pancreatic microsomes (CRM, 1 'equivalent' per 10 μl translation, as defined in *Walter and Blobel, 1983*), or microsomal membranes derived from Sf21 insect cells that contained an equivalent amount of recombinant Sec61 complex. Translation was initiated by transferring the reactions to 32°C for 30 min and stopped by removing the reactions to ice.

To isolate ER microsome-targeted RNCs, translation reactions were diluted with an equal volume of ice-cold high salt buffer (1M KOAc, 10 mM Mg(OAc)$_2$, 50 mM Hepes, pH 7.8) and sedimented through a high salt sucrose cushion (0.5 M KOAc, 0.5 M sucrose, 5 mM Mg(OAc)$_2$, 50 mM Hepes, pH 7.8) at 4°C in a TLA100 rotor (Beckman) for 10 min at 50,000 rpm. The membrane pellet was re-suspended to the original volume in membrane buffer (100 mM KOAc, 250 mM sucrose, 2 mM Mg(OAc)$_2$, 0.1 mM TCEP, 50 mM Hepes, pH 7.8).

For analysis of tRNA-associated nascent chains following protease digestion (as in *Figure 2—figure supplement 1*), reactions were quenched with PMSF as previously described (*Sharma et al., 2010*) and then rapidly diluted into 10 volumes of boiling 1% SDS, 0.1 M Tris, pH 6.8. Samples were then resolved on 4–12% NuPAGE Bis/Tris gels (Invitrogen, Carlsbad, CA), which run at neutral pH and therefore help preserve the alkaline-sensitive peptidyl-tRNA bond.

## Bis-maleimide crosslinking

ER microsome-targeted RNCs in membrane buffer were isolated as described above and treated with 50 μM bis-maleimidohexane (BMH, Pierce, Rockford, IL) or bis-maleimidoethane (BMOE, Pierce) for 30 min

at 0°C. To analyze total crosslinked products, reactions were quenched by addition of an equal volume of 2 × Laemmli sample buffer (which contains a large molar excess of DTT over BMH or BMOE), heated for 1 min at 95°C, and analyzed by SDS-PAGE and autoradiography. To immunoprecipitate (IP) proteins from the reactions, the samples were quenched with 1 mM DTT for 10 min at 0°C, denatured with 1% SDS at 95°C for 1 min, and diluted 10-fold with IP buffer (1% Triton-X 100, 100 mM NaCl, 50 mM Hepes, pH 7.8). Anti-FLAG affinity resin or protein A sepharose (GE Healthcare) along with the appropriate primary antibodies were then added and the samples were rotated overnight (~12 to 16 hr) at 4°C. The resin was washed four times with IP buffer supplemented with 0.1% SDS, and bound proteins were eluted with 1 × Laemmli sample buffer at 95°C for 1 min. Eluted material was analyzed by SDS-PAGE and autoradiography.

## NEM accessibility

ER microsome-targeted RNCs in membrane buffer were divided into two equal portions (10 μl each) and incubated with either 200 μM N-ethylmaleimide (NEM, Sigma Aldrich) or an equal volume of DMSO for 1 hr at 0°C. NEM was then quenched with 100 μl of quench buffer (5 mM DTT, 100 mM NaCl, 2 mM Mg(OAc)$_2$, 50 mM Hepes, pH 7.8) for 10 min at 0°C, and the membranes were isolated by centrifugation at 4°C in a TLA100 rotor (Beckman) for 10 min at 70,000 rpm. The membrane pellet was then solubilized in a detergent-containing buffer (1% SDS, 0.25 mM TCEP, 50 mM Hepes, pH 7.8) and treated with an equal volume (10 μl) of 16 mM 5 kDa PEG-maleimide (Nektar, San Francisco, CA), prepared in 50 mM Hepes (pH 7.8). Reactions were incubated at 32°C for 1 hr and quenched with 20 mM DTT for 20 min at 32°C. To completely hydrolyze the nascent chain-tRNA bond prior to SDS-PAGE (which simplified the appearance of autoradiograms), samples were treated with 100 μl of 200 mM Na$_2$CO$_3$ (pH 12) for 30 min at room temperature, then diluted with 1 ml of IP buffer (1% TX-100, 100 mM NaCl, 50 mM Hepes, pH 7.8), and precipitated with 10% trichloroacetic acid (TCA). Precipitated proteins were washed twice with ice-cold acetone, dissolved in 1 × Laemmli sample buffer and resolved by SDS-PAGE. The quantities of unmodified nascent chains (NC) and PEG-modified nascent chains (NC × PEG) were determined by phosphorimaging. The fraction of PEG-modified chains was defined as NC × PEG/(NC × PEG + unmodified NC). These values were normalized to control reactions that contained no NEM. NEM accessibility was defined as 1 − normalized fraction of PEG-modified chains.

## Photo-affinity labeling

Sf21 microsomes containing 50 nM Sec61 were treated with either 10 μM CT9 (*Maifeld et al., 2011*) or DMSO for 30 min at 0°C, followed by incubation with 50 nM CT7 for an additional 30 min at 0°C. Samples were then photolyzed and crosslinked proteins were detected by click chemistry, SDS-PAGE, and in-gel fluorescent scanning as previously described (*MacKinnon et al., 2007*; *MacKinnon and Taunton, 2009*).

## Sf21 cell culture, protein expression and purification of microsomes

Sf21 insect cells were grown and maintained in SF-900 II serum-free media (Gibco) at 27°C following standard protocols (*Fitzgerald et al., 2006*). The cells were transfected with the MultiBac vector using Fugene HD transfection reagent according to the manufacturer's instructions (Roche), and virus was propagated following published methods (*Fitzgerald et al., 2006*). For expression of Sec61α/γ complexes, a 100 ml culture of cells at $0.5 × 10^6$ cells/ml was infected with 2 ml of first generation virus (V$_1$), which resulted in immediate arrest of cell growth. The cells were harvested 48 hr after growth-arrest by sedimentation at 800×$g$ for 5 min. The cell were swollen in 20 ml hypotonic buffer (20 mM Hepes, pH 7.8) at 0°C for 20 min and broken with a microfluidizer (Emulsiflex-C5) at 15,000 psi for 10 min. The lysate was immediately adjusted to 100 mM KOAc, 5 mM Mg(OAc)$_2$, 1 mM EDTA, and 1 × EDTA-free protease inhibitor cocktail (Roche) and then clarified by centrifugation at 1000×$g$ for 10 min. To isolate microsomal membranes, the clarified lysate was centrifuged at 45,000 rpm in a type 70 Ti rotor (Beckman) for 1 hr at 4°C and the resulting microsomal pellet was re-suspended with a glass dounce in 400 μl of buffer (50 mM Hepes, 250 mM sucrose, 1 mM CaCl$_2$, pH 7.8). To remove endogenous RNA that co-purified with the membranes, the microsomes were treated with micrococcal nuclease (New England Biolabs, 150 units/ml final concentration) at 25°C for 10 min. The nuclease activity was quenched by adjusting the extract to 2 mM EGTA. Microsomes containing various Sec61 mutants were normalized for total protein by BCA protein assay (Pierce) and equivalent amounts of total protein resolved by SDS-PAGE next to a serial dilution of canine rough microsomes (CRM) containing a known concentration of the Sec61 complex. Proteins were then transferred to nitrocellulose and western blotted for the FLAG

epitope, Sec61α, and Sec61γ. Following this standardized protocol, the expression level between different Sec61α mutants was found to be very similar and the concentration of recombinant FLAG-Sec61 in the final microsomal preparation was similar to the concentration of Sec61 in CRM.

## Immunopurification of RNC-Sec61 complexes

Translation reactions (100 µl) were programed with mRNA templates encoding an N-terminal 3x-FLAG tag or 3x-HA tag (control) followed by the first 126 residues of TNFα. Reactions were assembled in the presence or absence of CT7 (1 µM). Following translation, ER microsome-targeted RNCs were isolated as described above, except the membrane pellet was brought up in two volumes of membrane buffer supplemented with 1% Deoxy BigChap (DBC, Anatrace, Santa Clara, CA). The membranes were solubilized for 10 min at 0°C and insoluble material removed by centrifugation at 50,000 rpm in a TLA100 rotor at 4°C for 10 min. The supernatant was then incubated with anti-FLAG affinity resin and mixed with rotation at 4°C for 2.5 hr. The resin was then sedimented (600 × g, 3 min, 4°C) and washed four times with 1 ml of membrane buffer containing 0.3% DBC. After the final wash, bound proteins were eluted with 250 µg/ml of 3x-FLAG peptide (Sigma) in membrane buffer containing 0.3% DBC. The eluted material was analyzed directly by SDS-PAGE and semi-quantitative western blotting, or was first photolyzed and then subjected to click chemistry and analyzed by SDS-PAGE and in-gel fluorescent scanning. Typically, ~0.3 pmol of purified RNC-Sec61 complexes were obtained from a 100 µl translation reaction.

## Cell culture, cell viability assay, and immunoblot analysis

HCT-116 cells and clonal lines were cultured in McCoy's 5A medium (Invitrogen) at 37°C with 5% $CO_2$. HEK293-FRT cells were grown in Dulbecco's Modified Eagle's Medium (Invitrogen) and grown at 37°C with 10% $CO_2$. All cultures were supplemented with 10% FBS and penicillin-streptomycin. Cell viability assays were performed by plating 2500 HCT-116 cells in flat-bottomed 96-well plates and treating with CT8 or CT9 the following day. After 72 hr, cell viability was analyzed using the Alamar Blue assay (Invitrogen) according to the manufacturer's instructions. HEK293 cells stably expressing wild-type or mutant Sec61α were generated using Flp-In 293 T-Rex cells (Invitrogen) as previously described (*Henise and Taunton, 2011*). For transient transfection experiments, $1.7 \times 10^5$ HEK293 T-Rex cells were seeded in 12-well dishes and transfected with a TNFα expression plasmid (*Maifeld et al., 2011*) using Lipofectamine 2000 (Invitrogen) for 2 hr, after which the culture media was changed to one containing increasing concentrations of CT8. After 24 hr, cells were harvested and lysed in 0.5% Triton X-100 in PBS. The resulting lysates were normalized using the Bradford assay (Bio-Rad) and 20 µg of total protein was resolved on 12% Tris-Tricine gels and analyzed by immunoblotting.

## Selection of resistant clones, RNA purification, RT-PCR and sequencing

To derive resistant cell lines, HCT-116 cells were incubated with 28–41 nM CT9 for 9–12 days, after which cell colonies were isolated by ring cloning and cultured in drug-free media. Total RNA was isolated from HCT116 cells using the RNeasy Mini Kit (Qiagen, Venlo, Netherlands) according to the manufacturer's instructions. Total cDNA was synthesized using anchored oligo-dT primers and Superscript III reverse transcriptase (Invitrogen) and *S61A1* cDNA was amplified with Phusion polymerase (Thermo Fisher Scientific) and sequenced bi-directionally by Sanger sequencing.

## Acknowledgements

The authors dedicate this article to the memory of Valerie Ohman. We thank Robert Weber for help in creating the stable cell lines. This work was supported by the US National Institutes of Health (GM081644 to JT), the Intramural Research Programs of the US National Institutes of Health and the UK Medical Research Council (RSH), the Academy of Finland (VOP), and the Sigrid Juselius Foundation (VOP).

## Additional information

### Funding

| Funder | Grant reference number | Author |
| --- | --- | --- |
| National Institutes of Health | GM081644 | Ramanujan S Hegde, Jack Taunton |
| Medical Research Council | | Ramanujan S Hegde |

| Funder | Grant reference number | Author |
|---|---|---|
| The Academy of Finland | | Ville O Paavilainen |
| Sigrid Juselius Foundation | | Ville O Paavilainen |

The funders had no role in study design, data collection and interpretation, or the decision to submit the work for publication.

## Author contributions

ALM, VOP, Conception and design, Acquisition of data, Analysis and interpretation of data, Drafting or revising the article; AS, Acquisition of data, Drafting or revising the article, Contributed unpublished essential data or reagents; RSH, JT, Conception and design, Analysis and interpretation of data, Drafting or revising the article

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
