## [Decision Letter]

Thank you for sending your work entitled “An allosteric Sec61 inhibitor traps nascent transmembrane helices at the lateral gate” for consideration at *eLife*. Your article has been favorably evaluated by a Senior editor and 3 reviewers.

The following individuals responsible for the peer review of your submission have agreed to reveal their identity: Randy Schekman and Reid Gilmore.

The Senior editor, Randy Schekman, and the three reviewers discussed their comments before we reached this decision, and the Senior editor has assembled the following comments to help you prepare a revised submission.

MacKinnon and colleagues have examined the mechanism of cotransin inhibition of the mammalian Sec61 complex. The authors present convincing evidence that cotransin stabilizes an early targeting intermediate. Elegant crosslinkng experiments localized the TM binding site to the cytosolic vestibule of Sec61. Point mutations in the TNF TM span reduce or enhance sensitivity to cotransin, depending upon whether the mutation increases hydrophobicity or alter the predicted helical propensity of the TM span. Cotransin insensitive mammalian cells were isolated, and found to have point mutations in the Sec61 plug domain. The most important feature of this manuscript is that it provides evidence for an initial binding site for signal sequences prior to insertion into the lateral gate.

The inhibitor is used to provide evidence for the existence of a transient pre-insertion intermediate but since this is based on the use of the inhibitor one may wonder about the general significance of this state. Several additional experiments are requested:

1) Figure 1: What is the exact evidence that the protease-protected fragment corresponds to the TMD plus the luminal C-terminal domain of TNFα. It seems to be based on an educated guess of the authors, but TMD regions run aberrantly on SDS-PAGE such that predicted mass on such gels may not be precise. The fragment might correspond to a ribosome-shielded peptide. Also, the fragment does not disappear completely after detergent solubilisation, which remains unexplained. The authors should demonstrate that the protected fragment contains the TM span, and also is a peptidyl-tRNA and whether Cys-46 or some of the other cysteine is part of the PK inaccessible fragment using the corresponding mutant TNFα.

2) Paragraph ‘CT8 traps the TMD helix in a defined orientation’: The authors argue that the CT8-arrested TMD/Sec61 complex would correspond to an otherwise transient, pre-integrated intermediate which is a selling point of the manuscript. However, this should be taken with caution since the experiments make use of a CT8-blocked translocon that is no longer responsive to the incoming nascent chain. It is unclear if the arrested state is truly a transient intermediate or the direct result of a block at Sec61, yielding an arrested non-physiological state where the translocon is unable to open because of its interactions with CT8. The authors argue that inhibition is allosteric, of which I am in complete agreement; however, this would suggest that the Sec61α/cotransin complex is in a distinct conformation relative to the channel alone. This would further suggest that the nature of the TMD interaction in the presence of cotransin does not necessarily reflect an on-pathway intermediate. In fact, the data might support a unique interaction as Figures 1 and 2 suggest that the nature of the interaction is distinct as CT8 enhances the pre-integration interaction (the authors note this at the end of the paragraph ‘CT8 stabilizes a transient pre-integrated intermediate’). This distinction is not critical to the story; however, it does qualify the ability to describe the cotransin complex as a 'pre-integrated intermediate'. In fact, the authors argue that a true pre-integrated TMD complex likely samples multiple conformations. So, at best, they are stabilizing one possible intermediate. In my opinion, they may be stabilizing a unique state that only appears in the presence of cotransin. A remedy might just be to provide some clarity to the 'intermediate' discussion early on.

3) Figure 1, Figure 1—figure supplement 1. It seems odd that the C49 appears to be 20% more accessible in the presence of CT8 when nascent chains are short (80 to 106 residues). This difference is small relative to that observed was observed with long nascent chains, so it does not raise concerns about the main conclusion. Do the authors have any explanation for this unexpected difference?

4) Figure 2. The authors state that the open and closed diamonds correspond to the NC-tRNA crosslinked to Sec61-α and Sec61-β. Identification of the major crosslinked bands was satisfactory (Figure 2—figure supplement 1). Have the authors shown that the NC-tRNA band and the open and closed diamond bands are base-labile as expected for peptidyl-tRNA? If so, they should add this statement to the text.

5) Figure 3 and Figure 3—figure supplement 1. The Results section states that the crosslinking pattern of V41P 126mers is different even though RNC targeting is not significantly affected. The crosslinking pattern is clearly different, but I am not sure how the authors reach the conclusion that RNC of V41P targeting was normal. The Sec61α crosslinks are similar in intensity (±CT8). Crosslinks seem to be elevated for -CT8 and reduced for +CT8. Sec61β crosslinks are mainly in the +CT8 lanes. In a later figure, data is presented that V41P 126mers progress to the integrated form based upon protease digestion, so it is not clear why any crosslinking is observed in the -CT8 case. Overall, I wonder whether the V41P mutation reduces integration at multiple points in the absence of CT8, and as such, whether analysis of this sample adds definitive information to Figure 3.

6) The observation that mutations in the plug region causes resistance to CT9 inhibition and reduced CT7 binding is interesting, but is no direct proof for a CT9 interaction site. The authors conclude that the mutant residues define the binding site for cotransin. While this interpretation is possible, an alternative interpretation is that point mutations in the plug domain indirectly perturb the cotransin binding site. The basis for this alternative interpretation is that there are a number of previously identified point mutations in the plug domain that are thought to destabilize the plug domain and allow more permissive translocation of precursors (the prl mutants). The most effective mutations identified in the current study (R66G and R66I) are probably prl alleles. R66 in human Sec61 corresponds to R67 in *S. cerevisiae* Sec61 and N65 in *E. coli* SecY. Martin Spiess isolated the R67E and R67C alleles, both of which have the prl phenotype. The *E. coli* N65Y mutant corresponds to prlA8914, which was isolated by the Beckwith lab. Rather it could be that these mutations effect a conformational change in Sec61-α that also reduces the effectiveness of CT9. In this respect, the probed sites cluster in a region that when mutated in the bacterial SecY causes prl mutations. While the Discussion tempers this, the description of the mutations in the results gives a strong impression that there is only one interpretation. It might be useful to include some other possible explanations. Certainly, with there being such a strong allosteric effect, one might imagine that mutations preventing this would be equally likely to appear. If one thinks broadly about the translocation channel, many mutations that affect specific function do not map to the location of function.

---

## [Author Response]

*1)*
Figure 1*: What is the exact evidence that the protease-protected fragment corresponds to the TMD plus the luminal C-terminal domain of TNFα. It seems to be based on an educated guess of the authors, but TMD regions run aberrantly on SDS-PAGE such that predicted mass on such gels may not be precise. The fragment might correspond to a ribosome-shielded peptide. Also, the fragment does not disappear completely after detergent solubilisation, which remains unexplained. The authors should demonstrate that the protected fragment contains the TM span, and also is a peptidyl-tRNA and whether Cys-46 or some of the other cysteine is part of the PK inaccessible fragment using the corresponding mutant TNFα*.

This is an important point, and our original statement about the identity of the protease- protected band in Figure 1 was indeed based on an educated guess. We have now performed an experiment to address the identity of this species (new Figure 1—figure supplement 1). We prepared stalled TNFα 126-mer RNCs similarly as in Figure 1 and treated them with proteinase K (PK). After SDS denaturation, we treated the samples with PEG-maleimide. PEG-Mal treatment resulted in a gel shift of the PK-protected nascent chain containing Cys49 in the TMD; no shift was observed with a Cys-free construct. The TNFα peptidyl- tRNA species was also labeled by PEG-Mal, which shows that the PK-protected band includes the C-terminus of the TNFα 126-mer (Figure 1—figure supplement 1).

The residual protease protection of the TNF 126-mer in the presence of detergent is likely the consequence of partial nascent chain shielding by the detergent solubilized ribosome/translocon complex (e.g., Mothes et al., JCB, 1998). This conclusion is supported by the observation that the fragment is not observed in the absence of microsomes, and is diminished when the TMD is prevented from fully engaging the Sec61 complex by CT8 (Figure 1, lanes 7 and 8).

*2) Paragraph ‘CT8 traps the TMD helix in a defined orientation’: The authors argue that the CT8-arrested TMD/Sec61 complex would correspond to an otherwise transient, pre-integrated intermediate which is a selling point of the manuscript. However, this should be taken with caution since the experiments make use of a CT8-blocked translocon that is no longer responsive to the incoming nascent chain. It is unclear if the arrested state is truly a transient intermediate or the direct result of a block at Sec61, yielding an arrested non-physiological state where the translocon is unable to open because of its interactions with CT8. The authors argue that inhibition is allosteric, of which I am in complete agreement; however, this would suggest that the Sec61α/cotransin complex is in a distinct conformation relative to the channel alone. This would further suggest that the nature of the TMD interaction in the presence of cotransin does not necessarily reflect an on-pathway intermediate. In fact, the data might support a unique interaction as*
Figures 1 and 2
*suggest that the nature of the interaction is distinct as CT8 enhances the pre-integration interaction (the authors note this at the end of the paragraph ‘CT8 stabilizes a transient pre-integrated intermediate’). This distinction is not critical to the story; however, it does qualify the ability to describe the cotransin complex as a 'pre-integrated intermediate'. In fact, the authors argue that a true pre-integrated TMD complex likely samples multiple conformations. So, at best, they are stabilizing one possible intermediate. In my opinion, they may be stabilizing a unique state that only appears in the presence of cotransin. A remedy might just be to provide some clarity to the 'intermediate' discussion early on*.

We agree that the precise structural relationship between the cotransin-stabilized state and the transient, pre-integrated intermediate is unclear. Nevertheless, our data clearly show: (1) in the absence of cotransin, the nascent TMD transiently occupies a configuration in which it crosslinks to Sec61α, (2) this state occurs prior to TMD integration, (3) this state (as defined by TMD crosslinking to Sec61α) is stabilized by cotransin (Figure 2), and (4) a ‘cotransin-resistant’ TMD mutant is transiently arrested in this state (by cotransin) prior to its escape and successful integration at longer nascent chain lengths (Figure 5). While it is possible that the cotransin-arrested Sec61 conformation is 'non-physiological', we feel it is more likely that cotransin binding stabilizes one of many conformational states normally accessed during TMD integration and prevents the transition of Sec61 to a fully open conformation. We have clarified this notion of the ‘cotransin-stabilized pre-integrated intermediate’ in the revised manuscript.

*3)*
Figure 1*,*
Figure 1—figure supplement 1*. It seems odd that the C49 appears to be 20% more accessible in the presence of CT8 when nascent chains are short (80 to 106 residues). This difference is small relative to that observed was observed with long nascent chains, so it does not raise concerns about the main conclusion. Do the authors have any explanation for this unexpected difference*?

We do not have an explanation for the observed difference.

*4)*
Figure 2*. The authors state that the open and closed diamonds correspond to the NC-tRNA crosslinked to Sec61-α and Sec61-β. Identification of the major crosslinked bands was satisfactory (*Figure 2—figure supplement 1*). Have the authors shown that the NC-tRNA band and the open and closed diamond bands are base-labile as expected for peptidyl-tRNA? If so, they should add this statement to the text*.

We have performed a new experiment to address this point (new Figure 2—figure supplement 1). The results show that the bands indicated in Figure 2 are sensitive to high pH and RNAse and thus are peptidyl-tRNA species. This experiment is now referenced in the main text.

*5)*
Figure 3
*and*
Figure 3—figure supplement 1*. The Results section states that the crosslinking pattern of V41P 126mers is different even though RNC targeting is not significantly affected. The crosslinking pattern is clearly different, but I am not sure how the authors reach the conclusion that RNC of V41P targeting was normal. The Sec61α crosslinks are similar in intensity (±CT8). Crosslinks seem to be elevated for -CT8 and reduced for +CT8. Sec61β crosslinks are mainly in the +CT8 lanes. In a later figure, data is presented that V41P 126mers progress to the integrated form based upon protease digestion, so it is not clear why any crosslinking is observed in the -CT8 case. Overall, I wonder whether the V41P mutation reduces integration at multiple points in the absence of CT8, and as such, whether analysis of this sample adds definitive information to*
Figure 3.

The V41P mutant integrates with similar efficiency as WT TNFα (Figure 6). Analysis of V41P 126-mers demonstrates cotransin-induced crosslinking to Sec61β, again with similar (albeit slightly reduced) efficiency as WT TNFα (Figure 3—figure supplement 1). By contrast, cotransin- induced crosslinking to Sec61α is much weaker with V41P than with WT 126-mers. Moreover, the helical periodicity observed with WT TNFα is not apparent with V41P. These observations support our contention that WT and T45L/T46L, but not V41P, adopt a helical conformation with a defined orientation in the cotransin-arrested state. Adopting this conformation and orientation is not prerequisite for successful integration or cotransin sensitivity.

*6) The observation that mutations in the plug region causes resistance to CT9 inhibition and reduced CT7 binding is interesting, but is no direct proof for a CT9 interaction site. The authors conclude that the mutant residues define the binding site for cotransin. While this interpretation is possible, an alternative interpretation is that point mutations in the plug domain indirectly perturb the cotransin binding site. The basis for this alternative interpretation is that there are a number of previously identified point mutations in the plug domain that are thought to destabilize the plug domain and allow more permissive translocation of precursors (the prl mutants). The most effective mutations identified in the current study (R66G and R66I) are probably prl alleles. R66 in human Sec61 corresponds to R67 in* S. cerevisiae *Sec61 and N65 in* E. coli *SecY. Martin Spiess isolated the R67E and R67C alleles, both of which have the prl phenotype. The* E. coli *N65Y mutant corresponds to prlA8914, which was isolated by the Beckwith lab. Rather it could be that these mutations effect a conformational change in Sec61-α that also reduces the effectiveness of CT9. In this respect, the probed sites cluster in a region that when mutated in the bacterial SecY causes prl mutations. While the Discussion tempers this, the description of the mutations in the results gives a strong impression that there is only one interpretation. It might be useful to include some other possible explanations. Certainly, with there being such a strong allosteric effect, one might imagine that mutations preventing this would be equally likely to appear. If one thinks broadly about the translocation channel, many mutations that affect specific function do not map to the location of function*.

We agree that the cotransin resistance mutations could potentially alter channel function. However, all of the identified resistance mutations cluster to the lumenal plug region; by contrast, prl mutations are distributed throughout the channel. Most importantly, we showed that select mutations (R66I and M136T) not only confer cotransin resistance in functional assays, but they also prevent CT7 binding. In our view, these combined observations indicate a high likelihood that the mutations are proximal to the cotransin binding site. We are unaware of examples where multiple drug resistance mutations cluster to the same site of the target and prevent drug binding, yet are located far away from the drug binding site. Nevertheless, such a scenario is possible and we have indicated this in the text.